# Cellular senescence in white matter microglia is induced during ageing in mice and exacerbates the neuroinflammatory phenotype

Tatsuyuki Matsudaira [1,7✉], Sosuke Nakano[1,7], Yusuke Konishi[1], Shimpei Kawamoto [1], Ken Uemura[1], Tamae Kondo[1], Koki Sakurai [2], Takaaki Ozawa [2], Takatoshi Hikida [2], Okiru Komine [3], Koji Yamanaka [3], Yuki Fujita[4], Toshihide Yamashita [4], Tomonori Matsumoto[1] & Eiji Hara [1,5,6✉]

Cellular senescence, a state of irreversible cell-cycle arrest caused by a variety of cellular stresses, is critically involved in age-related tissue dysfunction in various organs. However, the features of cells in the central nervous system that undergo senescence and their role in neural impairment are not well understood as yet. Here, through comprehensive investigations utilising single-cell transcriptome analysis and various mouse models, we show that microglia, particularly in the white matter, undergo cellular senescence in the brain and spinal cord during ageing and in disease models involving demyelination. Microglial senescence is predominantly detected in disease-associated microglia, which appear in ageing and neuro-degenerative diseases. We also find that commensal bacteria promote the accumulation of senescent microglia and disease-associated microglia during ageing. Furthermore, knockout of $p16^{INK4a}$, a key senescence inducer, ameliorates the neuroinflammatory phenotype in damaged spinal cords in mice. These results advance our understanding of the role of cellular senescence in the central nervous system and open up possibilities for the treatment of age-related neural disorders.

[1] Research Institute for Microbial Diseases (RIMD), Osaka University, 3-1 Yamadaoka, Suita, Osaka 565-0871, Japan. [2] Laboratory for Advanced Brain Functions, Institute for Protein Research, Osaka University, 3-2 Yamadaoka, Suita, Osaka 565-0871, Japan. [3] Department of Neuroscience and Pathobiology, Research Institute of Environmental Medicine, Nagoya University, Furo-cho, Chikusa-ku, Nagoya 464-8601, Japan. [4] Department of Molecular Neuroscience, Graduate School of Medicine, Osaka University, 2-2 Yamadaoka, Suita, Osaka 565-0871, Japan. [5] Immunology Frontier Research Center (IFReC), Osaka University, 3-1 Yamadaoka, Suita, Osaka 565-0871, Japan. [6] Center for Infectious Diseases Education and Research (CiDER), Osaka University, 2-8 Yamadaoka, Suita, Osaka 565-0871, Japan. [7] These authors contributed equally: Tatsuyuki Matsudaira and Sosuke Nakano. ✉email: matsudaira@biken.osaka-u.ac.jp; ehara@biken.osaka-u.ac.jp

While recent medical advances have increased life expectancy, many people suffer from age-related neural dysfunction and neurodegenerative diseases. Microglia, tissue-resident macrophages in the central nervous system (CNS), play important roles in maintaining homoeostasis by regulating neuronal activity and phagocytosing unwanted substances[1–4]. Recently, disease-associated microglia (DAM) with specific transcriptional signatures have been detected in the brains of elderly individuals, patients with various neurodegenerative diseases and neurodegenerative disease mouse models[5–9], suggesting DAM's involvement in age-related neural disease pathogenesis[1,3]. Furthermore, a recent analysis of the enhancer-promoter interactome using human brain cells revealed that variants related to sporadic Alzheimer's disease (AD) were primarily restricted to microglia enhancers[10]. Therefore, microglial abnormalities may be critically involved in degenerative CNS diseases.

Accumulating evidence indicates that cellular senescence leads to a decline in tissue function, associated with ageing and various diseases[11,12]. Cellular senescence is defined as irreversible cell-cycle arrest caused by a persistent DNA damage response, activated by various cellular stresses, such as telomere shortening, oncogene activation, excessive oxidative stress and radiation[13,14]. In addition to irreversible cell-cycle arrest, mainly coordinated by p16$^{INK4a}$ and p21$^{Waf1/Cip1}$, senescent cells exhibit a secretory phenotype that releases various inflammatory cytokines, chemokines and growth factors into their extracellular fluid[15–18]. This phenotype, now called senescence-associated secretory phenotype (SASP), plays beneficial and detrimental roles, depending on the biological context[14,19]. The accumulation of senescent cells can cause localised inflammation in the surrounding tissues, leading to cancer promotion and tissue dysfunction[20–22]. Furthermore, senescent-like signatures in the context of neurodegenerative diseases, including AD, multiple sclerosis (MS), Parkinson's disease, and amyotrophic lateral sclerosis, have been detected in various cell types in the CNS, including neural cells[23–25], cerebrovascular cells[26], and glial cells[23,27–32]. Importantly, removing senescent cells or inhibiting their accumulation reduces neuroinflammation and ameliorates cognitive functions[27–30,33,34], demonstrating that cellular senescence may contribute to CNS tissue dysfunction. In contrast, a comprehensive study by single-cell analysis of the hippocampus of aged mice revealed that p16-positive senescent cells primarily accumulated in oligodendrocyte progenitor cells (OPCs) and microglia during the natural ageing process[35]. Moreover, single-cell analysis in mice that can detect and remove p16-expressing cells[36] showed that microglia is one of the main populations expressing p16 in the aged brain[37], suggesting that glial cells, including microglia, are major cell types that experience senescence during the natural aging process. However, little is known about the regional specificity of senescent cell accumulation, mechanism of accumulation, or the outcome of the cell type-specific inhibition of senescent cells in the CNS.

In this study, we began by confirming senescent cells in the CNS during natural aging processes in detail and investigate their role in neural pathogenesis. We demonstrated that senescent cells accumulate in both the brain and spinal cord with age using p16 reporter (*p16-luc*) mice[38], which can help visualise senescent cells in vivo. Subsequent single-cell RNA sequencing (scRNA-seq) and detailed histological analysis revealed that senescent cells were primarily detected in microglia in the CNS white matter regions. Microglial senescence was prominently identified in DAM and further increased during ageing and in pathological conditions involving demyelination and neurodegeneration. Furthermore, we found that microglial senescence suppression attenuated the neuroinflammatory phenotype, and commensal bacteria partially promoted microglial senescence. These findings provide further insights into CNS cellular senescence and shed light on the role of microglial senescence in age-related and neurodegenerative CNS impairment progression.

## Results

**Senescent microglia expressing p16$^{INK4a}$ accumulate in the white matter of the brains of old mice.** To determine whether senescent cells accumulate in the ageing brain, we first analysed young (2–3 months old) and old (over 18 months old) *p16-luc* mouse brains, in which the expression of *p16$^{INK4a}$*, a key cellular senescence inducer, can be visualised as a bioluminescence signal[38]. Old *p16-luc* mice exhibited a marked increase in bioluminescence signal, especially in the pons and medulla oblongata regions, compared to young mice (Fig. 1a, b, Supplementary Fig. 1a). Further evaluation by immunoblotting and RT-qPCR revealed that *p16$^{INK4a}$* mRNA and protein were highly expressed in the brain's white matter regions, such as the corpus callosum and medulla oblongata in old mice (Fig. 1c, d). Furthermore, the expression of SASP factor genes, including *Il1b* and *Cxcl10*, was also increased, especially in the old mice's brain regions where *p16$^{INK4a}$* was highly expressed (Supplementary Fig. 1b, c). p16$^{INK4a}$ immunohistochemical staining using a specific antibody (Supplementary Fig. 1d, e) confirmed that the p16$^{INK4a}$-expressing cells were predominantly accumulated in the white matter of old mice but not in that of young mice (Fig. 1e, Supplementary Fig. 1f, g).

To characterise p16$^{INK4a}$-expressing cells, the corpus callosum was collected from young and old mice and subjected to scRNA-seq analysis (Fig. 2a). Cell-type classification based on their specific markers[39] and their uniform manifold approximation and projection (UMAP) plots (young, 4,657 cells; old, 6,789 cells) revealed that cells expressing *Cdkn2a*, encoding *p16$^{INK4a}$*, were abundant in the white matter of old mice, mainly in the microglia (Fig. 2b–e, Supplementary Fig. 2a–h), which is consistent with previous scRNA-seq analyses showing high p16 expression in microglia from the hippocampus or the whole brain of old mice[35,37]. Consistent with this data, microglia sorted from the old mice's brains showed higher *p16$^{INK4a}$* mRNA and protein expression than those of young mice (Fig. 2f, g, Supplementary Fig. 3a–c). Moreover, immunostaining analysis confirmed that the p16$^{INK4a}$-expressing cells were positive for Iba1, a microglial marker (Fig. 2h, i). These results, together with the observation that the γH2AX levels (another cellular senescence marker) increased in the old mice's microglia (Supplementary Fig. 3d), indicated that cellular senescence is induced in the microglia in the white matter during ageing.

Previous studies have demonstrated that both high p16 expression in microglia in the brain of old mice[35,37] and cell-cycle arrest are key features that prove cell senesce. To confirm that p16$^{INK4a}$-expressing microglia were irreversibly arrested in the cell cycle, we analysed 5-ethynyl-2'-deoxyuridine (EdU) uptake in the medulla oblongata of young and old mouse brains (Supplementary Fig. 4a). As reported previously, the number of microglia was increased in old mice[40]. Despite increased p16$^{INK4a}$-positive cells in the microglia, a significantly higher microglia percentage in old mice showed EdU incorporation than that in young mice (Fig. 2j, k, Supplementary Fig. 4b-d). However, p16$^{INK4a}$-positive microglia in old mouse brains incorporated much less EdU than p16$^{INK4a}$-negative microglia. Although EdU can be incorporated during DNA damage repair, and EdU in itself has been noted to be cytotoxic[41], this data suggests that the cell cycle was arrested in p16$^{INK4a}$-expressing microglia in old mice (Fig. 2l). Forced microglial proliferation experiments, in which PLX3397 (a Csf1r inhibitor) supplemented

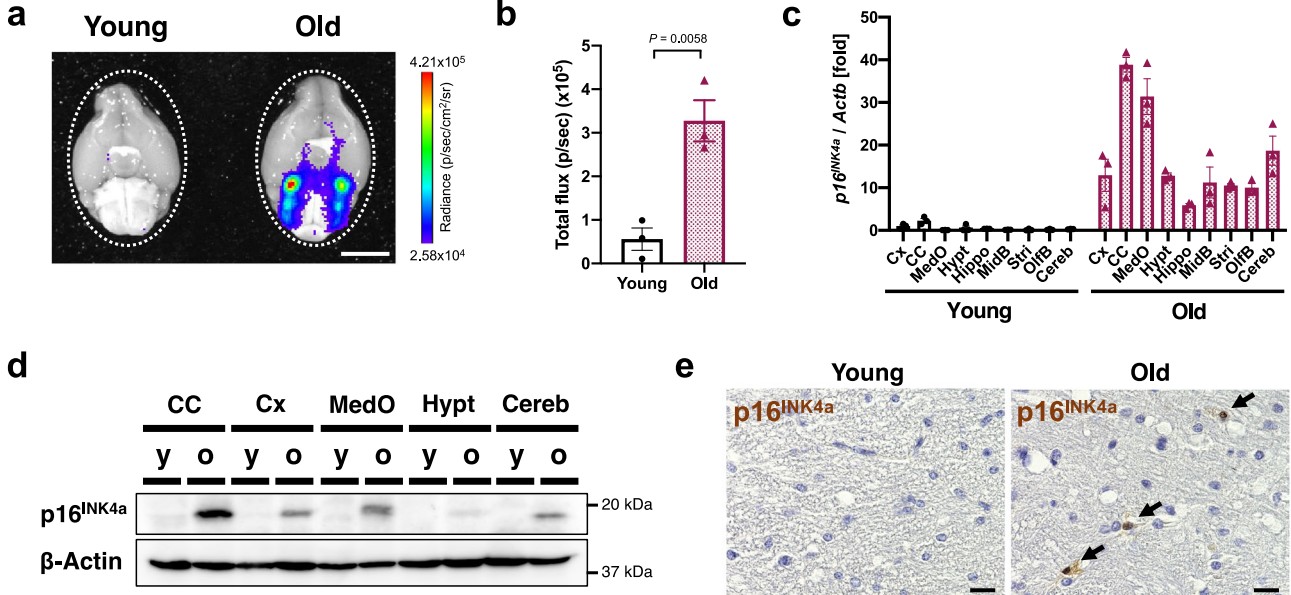

**Fig. 1 p16^INK4a positive senescent cells accumulate in white matter of the old mouse brain. a** Representative images depicting bioluminescence imaging of brains (ventral view) from young (2–3 months) and old (23–28 months) female *p16-luc* mice. The colour bar indicates radiance values, with minimum and maximum thresholds. **b** Bioluminescence intensity emitted from young and old mouse brains, given as total flux measured over the whole brain (region indicated with dotted line in (**a**)). **c** RT-qPCR analysis of *p16*^INK4a mRNA expression in lysates of each brain region from young (2 months) or old (33 months) male mice. *Actb* was used as a loading control. **d** Lysates of each brain region from young (2 months) or old (31 months) male mice were immunoblotted for p16^INK4a protein. β-Actin was used as a loading control. **e** Immunohistochemistry of p16^INK4a protein in the corpus callosum of young (2 months) and old (33 months) male mouse brains. Nuclei were stained with hematoxylin. Data presented as mean ± S.E.M of three brains from young or old mice (for **b, c**). Statistical significance was determined with two-tailed unpaired Student's *t*-test (**b**). Cx cortex, CC corpus callosum, MedO medulla oblongata, Hypt hypothalamus, Hippo hippocampus, MidB midbrain, Stri striatum, OlfB olfactory bulb, Cereb cerebellum, y young, o- old. Scale bar, 5 mm (for **a**) or 10 μm (for **e**).

the diet[42], induced transient depletion and subsequent microglia repopulation. This further confirmed that the number of EdU-positive microglia during repopulation was significantly lower in p16^INK4a-positive microglia than in p16^INK4a-negative microglia (Supplementary Fig. 4e–g). These results indicated that p16^INK4a-positive microglia are senescent and accumulate in the brain during ageing.

**p16^INK4a-expressing microglia are primarily DAM.** To further examine the p16^INK4a-positive microglial cell characteristics, we re-clustered the microglial populations in the scRNA-seq data. The composition of the major microglia subset in the old mice showed a different pattern from that in the young mice (Fig. 3a–c). Most microglia were primarily separated into three cluster groups characterised by different sets of transcription factors: clusters expressing homoeostatic genes (Clusters 0, 1 and 3), inflammatory cytokines (Clusters 2, 4, 5 and 7), or DAM-related genes (Clusters 6 and 8)[5,7] (Fig. 3d, e, Supplementary Fig. 5a–f). Other small clusters (ribosome-related cluster 10 and unspecified cluster 9) were also identified (Fig. 3c, d). Amongst the three cluster groups, *Cdkn2a* was particularly highly expressed in the DAM-related cluster (clusters 6 and 8), observed only in old mice (Fig. 3f–h), as is the case with the microglia in the hippocampus of APP/PS1 AD model mice[32]. Genes associated with ageing or cellular senescence[32] were also strongly expressed in the DAM-related cluster (Supplementary Fig. 5g), indicating that microglia expressing DAM-related genes have a senescent signature in the white matter of old mice. Importantly, although the microglia were reported to be activated ex vivo during enzymatic tissue digestion[43], genes related with ex vivo activated microglia (exAM) were relatively low in the DAM-related cluster, suggesting that DAM-related genes were not caused by enzymatic activation during cell isolation (Supplementary Fig. 5h). Pseudotime

analysis of single-cell trajectories in Monocle 2 showed that each cluster was connected by a curve, suggesting that DAM-related clusters are the most developed and that microglial senescence is dependent on the development of a microglial state (Supplementary Fig. 5i–l).

A previous study demonstrated an inverse correlation between the surface protein expression of a DAM marker Cd11c (Itgax) and telomere length in senescent microglia in APP/PS1 mice[32]. To further verify the relationship between p16^INK4a and DAM, microglia positive for Cd11c (Itgax), a DAM marker, were sorted from old mouse brains using FACS (Supplementary Fig. 6). Cd11c-positive microglia highly expressed *p16*^INK4a in addition to various DAM-related genes, including *Cd11c*, *Apoe* and *Axl*, compared with Cd11c-negative microglia (Fig. 3i). Immunostaining analysis also confirmed that p16^INK4a and Lgals3, another DAM marker, were frequently co-expressed in Iba1-positive microglia (Fig. 3j, k). Over 70% of Lgals3-positive DAM were positive for p16^INK4a in the corpus callosum of old mice (Fig. 3l). These results indicated that p16^INK4a-positive senescent microglia, which accumulate in the brain with age, exhibit features associated with DAM, and most DAM are senescent in the old mice's brains.

**Commensal bacteria facilitate the accumulation of senescent microglia and senescent DAM.** Next, we investigated how cellular senescence is triggered in microglia during ageing. We have previously reported that gut bacteria promote the induction of cellular senescence in various organs, such as the liver and intestine in vivo[20,22]. Gut bacteria also affect transcriptional states and increase cellular reactive oxygen species production in microglia through various metabolites[44–46]. Therefore, we wondered whether microglial cellular senescence in ageing is affected by commensal

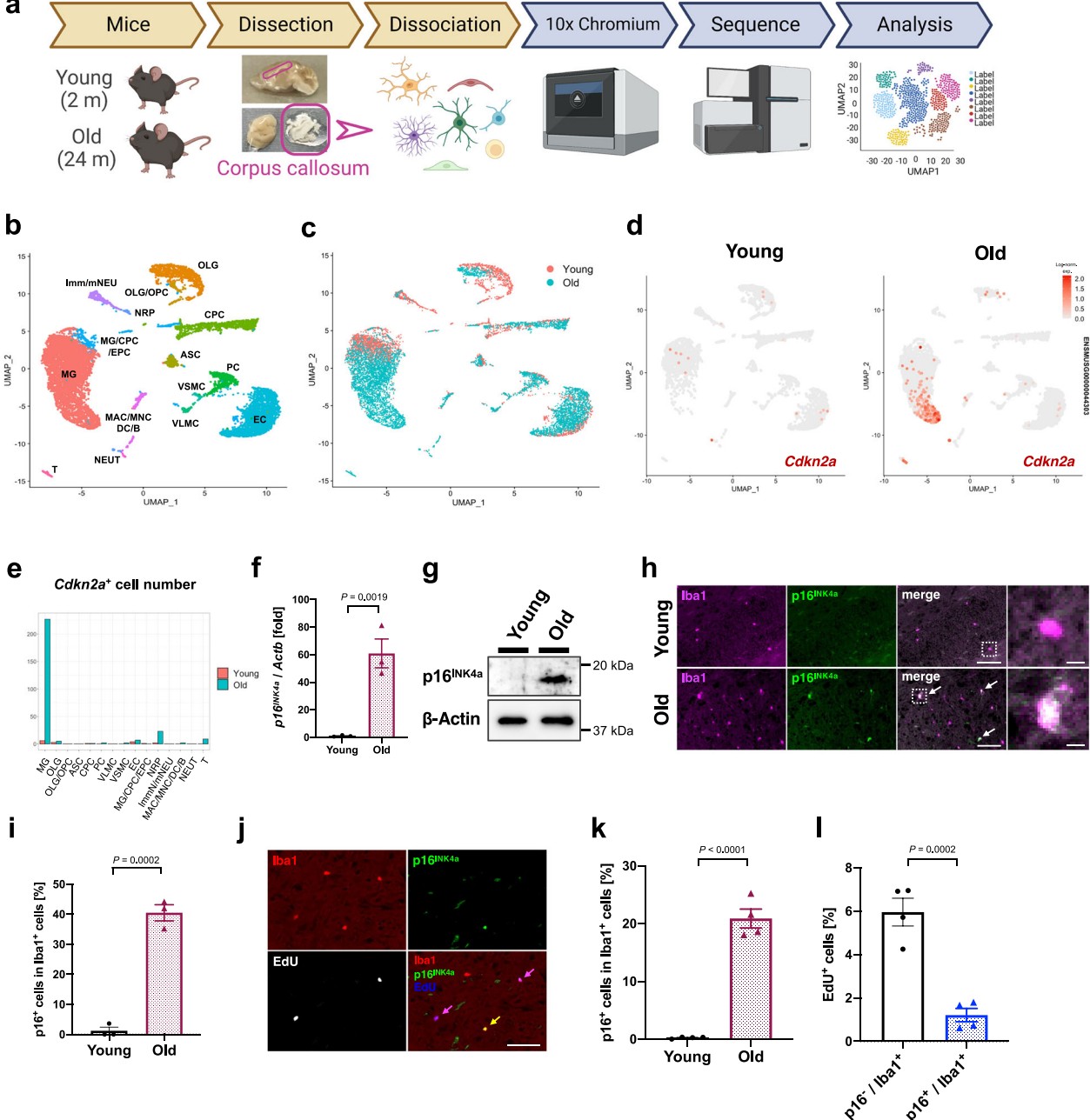

**Fig. 2 p16^INK4a positive senescent microglia accumulate in white matter of the old mouse brain. a** Schematic depicting the workflow for single-cell RNA-sequencing of the corpus callosum in young (2 months) and old (24 months) male mice. The figure was created with BioRender.com. **b** Single-cell RNA transcriptomic analysis of corpus callosum (CC) from young and old male mice. Uniform manifold approximation and projection (UMAP) plot of 11,446 cells (young, N = 1, n = 4,657 cells; old, N = 1, n = 6,789 cells) was analysed via Seurat and labelled using cell type-specific markers. Each dot represents a single cell. **c** UMAP plot of cells annotated by each sample (young CC, red; old CC, blue) **d** UMAP plot depicting the expression kinetics of *Cdkn2a* in the young and old CC. **e** Distribution of *Cdkn2a*-positive cells in (**d**) displayed by cell type. **f** RT-qPCR analysis of *p16^INK4a* mRNA expression in lysates of Cd11b+ and Cx3cr1+ cells from the brains of young (2 months, N = 3) and old (24 months, N = 3) male mice. *Actb* was used as an internal control. **g** Lysates of MACS-isolated Cd11b+ cells from young (2 months) and old (33 months) male mouse brains were immunoblotted for p16^INK4a protein. β-Actin was used as a loading control. **h** Representative immunofluorescence images for Iba1 (magenta) and p16^INK4a (green) in the CC of young (2 months) and old (33 months) male mouse brains. Magnified images of boxed areas are shown in the right column. Colocalisation of Iba1 with p16^INK4a is indicated by arrows. **i** Quantification of p16^INK4a expression amongst Iba1+ microglia in (**h**) (young: N = 3, n = 81–103 Iba1+ cells, old: N = 3, n = 238–394 Iba1+ cells per mouse). **j** Representative immunofluorescence images for Iba1 and p16^INK4a in the medulla oblongata of young (2 months) or old (28 months) male mouse brain. EdU was detected using Click-iT EdU cell proliferation kit. Iba1+/p16+/EdU- cells (yellow) and Iba1+/p16-/EdU+ cells (magenta) are indicated by arrows. **k** Quantification of p16^INK4a expression amongst Iba1+ cells in (**j**) (young: N = 4, n = 157–368 Iba1+ cells; old: N = 4, n = 514–747 Iba1+ cells per mouse). **l** Quantification of EdU-positive cell ratio amongst p16-/Iba1+ cells or p16+/Iba1+ cells in the medulla oblongata of old male mice, related to (**j**) (old: n = 403–612 p16-/Iba1+ cells or 111–161 p16+/Iba1+ cells out of 514–747 Iba1+ cells per mouse). Data presented as mean ± S.E.M. Statistical significance was determined with two-tailed unpaired Student's t-test (**f, i, k, l**). Scale bar, 50 μm (for **h, j**) or 5 μm (for **h**, magnified images). Log-norm. exp. Log-normalised expression.

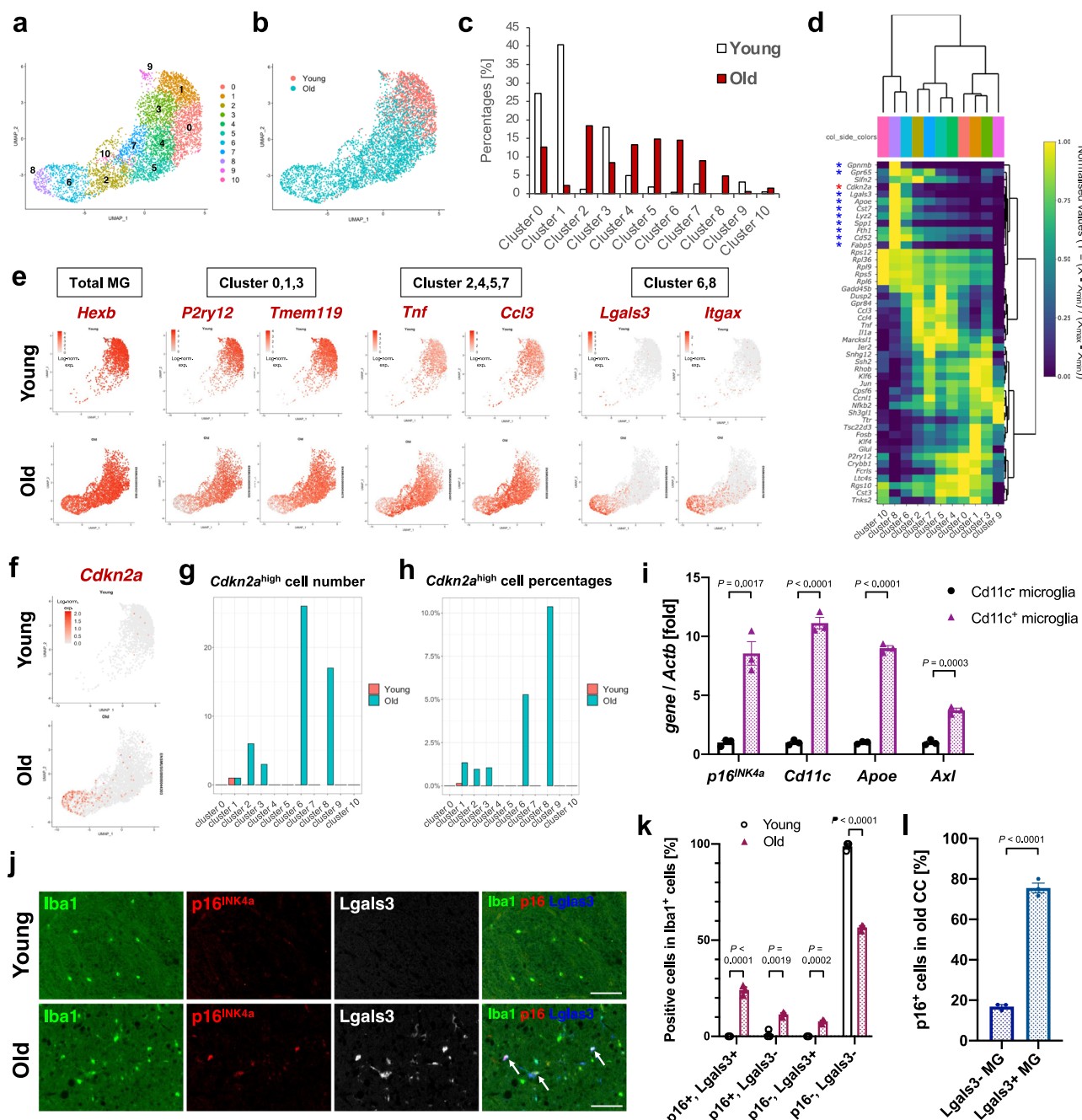

**Fig. 3 p16INK4a positive senescent microglia highly expressed DAM markers. a** UMAP plot of 5,205 individual microglia in the corpus callosum (CC) of young (2 months, $N = 1$, $n = 1,792$ cells) or old (24 months, $N = 1$, $n = 3,413$ cells) male mice. Each dot represents a single cell. Numbers indicate identified clusters. **b** UMAP plot of microglia annotated by each sample (young CC, red; old CC, blue) **c** Proportion of microglia belonging to each cluster in (**a**) between young and old mice. **d** Heatmap of genes including *Cdkn2a* and top 5 genes identified by model-based analysis of single cell transcriptomics (MAST) in each cluster. Normalised values per each cluster in each gene are shown. DAM-related genes and *Cdkn2a* are marked with blue or red asterisks, respectively. **e** UMAP plots depicting expression patterns of the microglia-specific *Hexb* gene as a marker of total microglia, and genes selectively expressed in cluster 0,1,3, cluster 2,4,5,7 or cluster 6,8. **f** UMAP plots depicting the expression patterns of *Cdkn2a* gene in microglia from young or old mouse brains. **g, h** Cell numbers (**g**) and proportion (**h**) of *Cdkn2a*-high cells (expression level ≥ 1) in each cluster. **i** RT-qPCR analysis of *p16INK4a*, *Cd11c*, *Apoe*, and *Axl* mRNA from lysates of Cd11b+/Cx3cr1+/Cd11c+ or Cd11b+/Cx3cr1+/Cd11c- microglial cells from old male mouse (28–29 months). *Actb* was used as an internal control. **j** Representative immunofluorescence images for Iba1 (green), p16INK4a (red), and Lgals3 (white or blue) in the corpus callosum of young (2 months) or old (33 months) male mice. Arrows indicate triple-positive (Iba1+/p16+/Lgals3+) cells. **k** Quantification of p16+/Lgals3+, p16+/Lgals3-, p16-/Lgals3+ and p16-/Lgals3- Iba1+ cells in (**j**) (young: $n = 81$–103 Iba1+ cells; old: $n = 151$–177 Iba1+ cells per mouse). **l** Quantification of p16INK4a expression amongst Iba1+/Lgals3- or Iba1+/Lgals3+ cells in (**k**). Data presented as mean ± S.E.M from three mice (for **i**, **k**, and **l**). Statistical significance was determined with two-tailed unpaired Student's *t*-test (**i**, **k**, **l**). Scale bar, 50 μm. MG microglia, CC corpus callosum. Log-norm. exp. Log-normalised expression.

bacteria. To test this hypothesis, we first compared $p16^{INK4a}$ expression in the medulla oblongata of the old mice, raised under specific pathogen-free (SPF) or germ-free (GF) conditions, and found that $p16^{INK4a}$ expression was significantly reduced in the medulla oblongata of old GF mice (Fig. 4a). To gain further insight into senescent microglia in these mice, we performed scRNA-seq analysis on cells freshly collected from the medulla oblongata of young and old mice raised under GF and SPF conditions (Supplementary Fig. 7a-e). Microglial population re-clustering in the scRNA-seq data (Fig. 4b-d) showed that *Cdkn2a*-positive microglia were primarily annotated in cluster 7 (a DAM-related cluster) in addition to other clusters (clusters 3, 4 and 8) (Fig. 4e, f). More cells with high expression of *Cdkn2a* or DAM-related genes were observed in old SPF mice than in old GF mice (Fig. 4g-j, Supplementary Fig. 7f-n). Consistently, immunostaining analysis revealed that the numbers of p16$^{INK4a}$-expressing cells, DAM and p16$^{INK4a}$-positive DAM were significantly lower in old GF mice than those in old SPF mice despite no change in the total number of microglia (Fig. 4k-p). Furthermore, bulk RNA-seq analysis performed on FACS-sorted microglia further supported that DAM- and cell cycle-related gene expressions were suppressed in the microglia of old GF mice compared to old SPF mice (Supplementary Fig. 8a-e). Collectively, these data suggest that commensal bacteria promote the induction of age-related DAM and CNS microglial senescence.

**Inhibition of microglial cellular senescence reduces neuropathy in a multiple sclerosis mouse model**. Next, we asked whether the findings observed in the brain could also occur in other CNS regions, such as the spinal cord. Analysis using *p16-luc* mice confirmed that the bioluminescence signal markedly increased in the old mice's spinal cords (Fig. 5a, b). Increased $p16^{INK4a}$ and *Il1b* (an inflammatory SASP factor) expression was also observed (Fig. 5c, d). Immunostaining revealed that in the spinal cord white matter of old mice, about half of the microglia were p16$^{INK4a}$-positive, and most ( ~ 90%) of the Lgals3-positive DAM were p16$^{INK4a}$-positive (Fig. 5e–g). Furthermore, senescent DAM accumulation in the spinal cord was observed in the paralysis phase in SOD1 G93A mice[47], recapitulating amyotrophic lateral sclerosis (ALS) pathogenesis (Supplementary Fig. 9a–e). These results indicated that microglial senescence occurs throughout the CNS under ageing and diseased conditions and is predominantly induced in the CNS white matter DAM.

We then investigated whether senescent microglia are involved in the pathogenesis of neurodegenerative diseases. To this end, mice were subjected to experimental autoimmune encephalomyelitis (EAE), a mouse model of MS, to induce spinal cord demyelination and limb paralysis[48]. Notably, p16$^{INK4a}$-positive microglia and Lgals3-positive DAM accumulated at the EAE peak, and most DAM expressed p16$^{INK4a}$ (Fig. 5h–j). To investigate the role of senescent microglia in the progression of spinal cord impairment, we utilised $p16^{INK4a}$ knockout (p16 KO) mice, in which the progression of cellular senescence was suppressed[49]. In contrast to WT mice, the degree of paralysis in EAE mice was significantly attenuated in p16 KO mice (Fig. 5k–m). In the spinal cords of p16 KO mice, Cd4$^{+}$ T cell infiltration was significantly reduced during the EAE acute phase (Supplementary Fig. 10a–k), as was the demyelination progression (Fig. 5n, o).

To further confirm the role of senescent microglia in neuroinflammatory pathogenesis, we generated microglia/macrophage-specific $p16^{INK4a}$ knockout mice using Cx3cr1-CreER$^{T2}$: p16$^{flox/flox}$ mice (MG-p16 KO mice) (Supplementary Fig. 11a). Significant alleviation of the EAE phenotype was observed in MG-p16 KO mice treated with tamoxifen, although not to the extent observed in systemic p16 KO mice (Fig. 5p-r). This difference may be due, at least in part, to the incomplete efficiency of $p16^{INK4a}$ knockout by

tamoxifen administration (Supplementary Fig. 11b-e). These results indicated that the EAE phenotype is attenuated by inhibiting microglial cellular senescence, suggesting the involvement of senescent microglia and DAM in neurodegenerative spinal cord diseases.

## Discussion

We demonstrated that microglia in the white matter represent the main cell type that undergoes cellular senescence in the CNS during ageing and EAE. Comprehensive investigations using various mouse models have revealed that microglial senescence occurs in the brain and spinal cord under ageing and disease conditions. Microglial senescence is predominantly induced in DAM (Figs. 3l, 4p, 5g, 5j, Supplementary Fig. 9e), which have been reported to be involved in neurodegenerative diseases[5,7]. Suppression of senescence induction by knockout of the $p16^{INK4a}$ gene in microglia ameliorated the neural impairment phenotype in the EAE model (Fig. 5p-r). We have also shown, by comparing GF and SPF mice, that the induction of DAM and cellular senescence is influenced by commensal bacteria (Supplementary Fig. 12).

Recent reports analysing microglia in bulk have shown increased expression of senescence markers, including p16$^{INK4a}$ and p21$^{Waf1/Cip1}$, in old mouse brains[50] or the ALS rat model spinal cord[51]. Moreover, scRNA-seq analysis of old mouse brains revealed that microglia are the primary cells expressing a reporter driven by the *p16* promoter[37]. The present study confirms that microglia undergo cellular senescence by showing high p16$^{INK4a}$ expression and cell-cycle arrest, further extending previous notions by elucidating that microglial senescence is induced in both the brain and spinal cord by ageing and disease. Although we only analysed one mouse brain per group in our scRNA-seq study, the induction of senescence in microglia was consistently confirmed in various experiments using a significant number of brains from young and old mice (Figs. 2f-k, 3i-l, 4l, 4n-p, Supplementary Fig. 3c). Notably, microglial senescence was mostly observed in the white matter DAM. In contrast, a previous scRNA-seq analysis by Talma et al. failed to detect the occurrence of senescent cells amongst microglial subclusters, perhaps due to the difference in sampling regions and the reporter sensitivity[37]. Safaiyan et al. recently reported white matter-associated microglia (WAM), which are microglia with parts of the DAM gene signature specifically observed in the brain white matter of old mice[52]. Although we could not find clusters expressing WAM-specific genes in our scRNA-seq analysis (Supplementary Fig. 13), further studies on the diversity of microglia in ageing and diseased brains are warranted.

The causal relationship between the microglial state transition into the DAM and cellular senescence is not straightforward. For example, Hu et al. reported that microglial proliferation is associated with DAM development and cellular senescence feature acquisition in APP/PS1 AD model mice[32]. They proposed that DAM are senescent microglia resulting from intense proliferation. However, our results indicate that although senescent microglia are present in DAM, not all DAM are senescent microglia; some senescent microglia are not classified as DAM (Figs. 3k, 4o, 5f, 5i, Supplementary Fig. 9d). It is therefore unlikely that the mechanisms that lead to DAM are the same as those that cause cellular senescence. Furthermore, blockade of cellular senescence did not significantly decrease the Cd11c-positive DAM number (Supplementary Fig. 10b). These findings suggest that common triggers can lead to a parallel transition from homoeostatic microglia to DAM and cellular senescence in ageing and diseased states. However, these two processes are independent of each other. We demonstrated that microglia/macrophage-specific $p16^{INK4a}$ knockout attenuated the neuroinflammatory

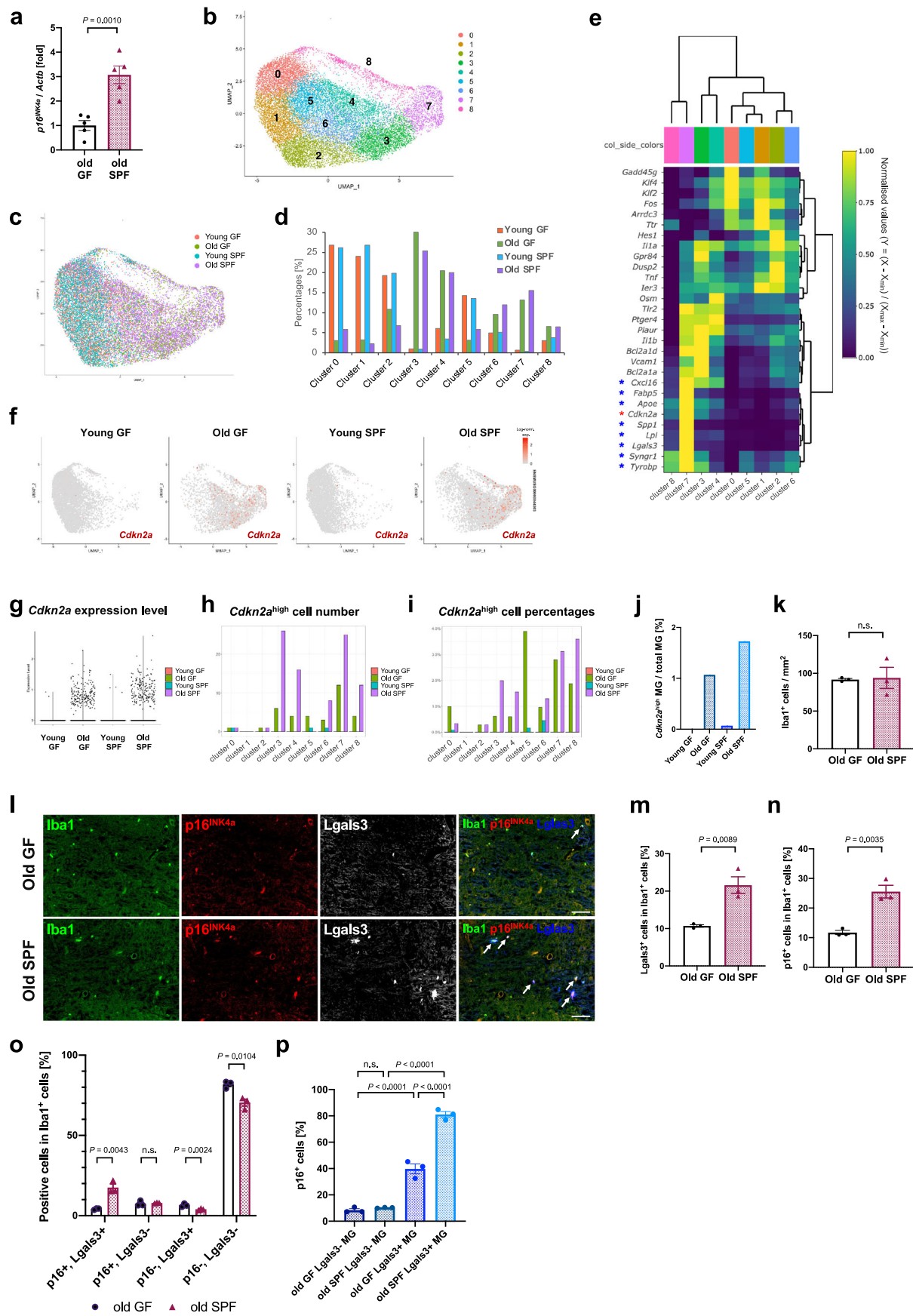

**Fig. 4 Commensal bacteria promote microglial senescence. a** RT-qPCR analysis of $p16^{INK4a}$ mRNA in lysates of medulla oblongata from old GF (20 months, $N = 5$) or SPF (20 months, $N = 5$) male mice. *Actb* was used as an internal control. **b** Re-clustered UMAP plot of 18,423 individual microglia in the medulla oblongata of male mice (young GF: 3 months old, $N = 1$, $n = 5620$ cells; old GF: 25 months old, $N = 1$, $n = 3276$ cells; young SPF: 3 months old, $N = 1$, $n = 4373$ cells; old SPF: 23 months old, $N = 1$, $n = 5154$ cells). Each dot represents a single microglia. Numbers indicate identified clusters. **c** UMAP plot of cells annotated by each sample (Young GF, red; Old GF, green; Young SPF, blue; Old SPF, purple). **d** Distribution of microglia from each group between clusters depicted in (**b**). **e** Heatmap of genes including *Cdkn2a* and top 5 genes identified by MAST in each cluster. Normalised values per each cluster in each gene are shown. DAM-related genes and *Cdkn2a* are marked with blue or red asterisks, respectively. **f** UMAP plot depicting expression of *Cdkn2a* in young/old GF or SPF mice. **g** Overall *Cdkn2a* expression level in young/old GF or SPF mice. Each dot represents a single cell. **h, i** Absolute numbers (**h**) and proportion (**i**) of *Cdkn2a*-high cells (expression level $\geq 1$) in each cluster. **j** *Cdkn2a*-high microglia (MG) as a percentage of total MG in (**g**). **k** Quantification of Iba1 positive cells per mm$^2$ in the medulla oblongata of old GF (20 months, $N = 3$, $n = 494$–$737$ Iba1$^+$ cells per mouse) or old SPF (20 months, $N = 3$, $n = 547$–$732$ Iba1$^+$ cells per mouse) male mice. **l** Representative immunofluorescence images for Iba1 (green), p16$^{INK4a}$ (red) and Lgals3 (white or blue) in the medulla oblongata of old GF or SPF male mice, related to (**k**). Arrows indicate triple-positive (Iba1$^+$/p16$^+$/Lgals3$^+$) cells. **m, n** Quantification from (**l**) of the number of Lgals3$^+$ cells (**m**) or p16$^+$ cells (**n**) as a percentage of total Iba1$^+$ cells. **o** Quantification from (**l**) of the percentages of p16$^+$/Lgals3$^+$, p16$^+$/Lgals3$^-$, p16$^-$/Lgals3$^+$ and p16$^-$/Lgals3$^-$ Iba1-positive cells. **p** Percentages of p16$^{INK4a}$ positive Iba1$^+$/Lgals3$^-$ or Iba1$^+$/Lgals3$^+$ cells, related to (**o**). Data presented as mean $\pm$ S.E.M. Statistical significance was determined with two-tailed unpaired Student's *t*-test (**a, k, m, n, o**) and one-way ANOVA followed by Tukey's multiple comparison test (**p**). n.s. non-significant. Scale bar, 50 μm. Log-norm. exp. Log-normalised expression.

phenotypes in the EAE model (Fig. 5p-r). $p16^{INK4a}$ deletion in bone marrow-derived macrophages is reported to lead to a reduction in pro-inflammatory signalling[53]. Moreover, microglia with high p16 promoter activity exhibited signatures associated with cytokine production and T cell infiltration[37]. Therefore, it is tempting to speculate that cellular senescence with high p16$^{INK4a}$ expression in microglia may be closely related to transcriptional polarisation, leading to inflammatory progression of the local niche. We found that there is a difference in phenotype intensity between systemic p16-KO mice and microglia/macrophage-specific p16-KO mice (Fig. 5k, p). Although the insufficient knockout of p16 could be one of the main reasons behind the aforementioned inflammatory progression (Supplementary Fig. 11b-e), it is tempting to consider other cell types that dramatically change their dynamics in the local niche in the process of EAE-induced demyelination, including oligodendrocytes, OPCs, and T-cells[54,55]. Interestingly, our data showed that the proportion of Cdkn2a-expressing T-cells was increased in the white matter of aged mice (Supplementary Fig. 2f), suggesting that these cells may be also senescent and contribute to the severity of the EAE phenotype.

From the perspective of preventing ageing-related CNS disorders, it is important to understand commensal bacterial involvement in the senescent microglia appearance in age-related CNS disorders and neurodegenerative diseases. Reduced oxidative stress and improved mitochondrial dysfunction were recently reported in the brain microglia of old GF mice compared to their SPF counterparts[46]. Moreover, GF mice and mice treated with antibiotics exhibited attenuated phenotypes in AD and MS models[56,57], although the microglial senescence involvement in these mechanisms is unknown. Consistent with these reports, our results suggest that commensal bacteria are involved in the induction of DAM and microglial senescence (Fig. 4g-j, l-p, Supplementary Figs. 7f-n, 8d, e), which is possibly associated with neural impairment pathogenesis. However, microbiota are also known to affect immune cells throughout the body, making it difficult to distinguish the effects of commensal bacteria on microglia and immune cells. Further studies are needed to clarify the relationship between commensal bacteria and age-related CNS impairment via microglial senescence.

Our results revealed the occurrence and characteristics of microglial senescence, as well as their possible precursive stimuli and role in neural impairments. These findings suggest possible strategies to suppress the development of age-related neural diseases, such as senotherapy[58,59], targeting senescent cells.

## Methods

**Animal experiments.** Young and old wild-type (WT) C57BL/6 mice were purchased from Charles River Laboratories Japan and CLEA Japan. The *p16-luc* mice and the EAE-induced mice were female, and the other mice were male. p16 knockout mice are reported previously[49]. Cx3cr1-CreER$^{T2}$ mice (B6.129P2(C)-Cx3cr1tm2.1(cre/ERT2)Jung/J)[60] were kindly provided by Prof. Dr. Toshihide Yamashita. p16$^{flox/flox}$ mice[61] were kindly provided by Prof. Dr. Norman E Sharpless. 100 ul of 20 mg/ml Tamoxifen dissolved in corn oil were intraperitonially injected daily for 5 days in Cx3cr1-CreER$^{T2}$: p16$^{flox/flox}$ mice and maintained 3–4 weeks before EAE experiments to reduce the effect of p16$^{INK4a}$-depleted lymphocytes such as monocytes and peripheral macrophages[60]. Albino *p16-luc* mice were generated by crossing with B6 Albino mice (Charles River Laboratories Japan) and used for bioluminescence imaging. The mice were bred and maintained under specific pathogen-free (SPF) conditions, at 23 °C ± 2 °C and 55% ± 15% humidity on a 12-h light–dark cycle, and fed normal diet (ND, CE-2 from CLEA Japan Inc., composed of 12 kcal% fat, 29 kcal% protein and 59 kcal% carbohydrates) ad libitum in SPF facility at the Research Institute for Microbial Diseases (RIMD), Osaka university. PLX3397 (Selleckchem, TX, USA, cat# S7818) was formulated at 0.029% in AIN-76A chow by Research Diets (NJ, USA). Mouse chow containing 0.25% w/w cuprizone (bis-cyclohexanone oxaldihydrazone, Merck Millipore, MA, USA, cat#: C9012) was custom-synthesised (Oriental Yeast Co. LTD., Japan). All animals were maintained according to the protocols approved by the Committee for the Use and Care of the RIMD. Albino *p16-luc* mice analysed by RNA-seq were sterilised and maintained in vinyl isolators in the animal facility at CLEA Japan and used as GF mice in this paper. The frozen brain samples of SOD1 G93A mice (Strain # 004435, Jackson laboratories, USA) were kindly provided by Prof. Dr. Koji Yamanaka. For irradiation experiments (Supplementary Figs. 1e and 11b), young mice (3 months) were anaesthetised by intraperitoneal injection of three anaesthetic agents (medetomidine (0.75 mg/kg, Zenoaq, Japan), midazolam (4 mg/kg, Sandoz, Japan) and butorphanol (5 mg/kg, Meiji Seika Pharma, Japan)). The mouse body under the head was carefully shielded by a lead collimator. Irradiated animals received γ-ray irradiation of the head with a total dose of 10 Gy with Gammacell 40 Exactor (Nordion, Canada) in the RIMD. Antisedan (0.75 mg/kg, Zenoaq) were then intraperitoneally injected after irradiation and the mice were analysed in 3 months.

**Flow cytometry analysis.** Mice were anaesthetised by intraperitoneal injection of three anaesthetic agents and heparin and transcardially perfused with ice-cold HBSS containing heparin. Isolated brains or spinal cords were roughly minced by scissors, followed by digestion with 0.1% Collagenase Type I (Fujifilm, Japan, cat#: 035-17604) containing 1 mM calcium chloride and 0.3 mM magnesium chloride for 30 min at 37 °C or digestion according to the standard protocol of Neural tissue dissociation kits (P) (Miltenyi Biotec, Germany, cat#: 130-092-628). The homogenates were filtered through a 70 μm cell strainer and the cell debris was removed by centrifugation according to the standard protocol of Debris Removal Solution (Miltenyi Biotec, Germany, cat#: 130-109-398). The cells in HBSS with 1% BSA were then treated with CD16/32 FcR-blocking reagent (Biolegend, CA, USA, cat#: 101330, clone: 93) on ice for 10 min, followed by staining with fluorescence-conjugated antibodies for 15 min. The following antibodies were used (all antibodies were purchased from BioLegend): fluorescein isothiocyanate (FITC)-conjugated anti-Cd11b (cat#: 101206, clone: M1/70), phycoerythrin (PE)-conjugated anti-Cd11c (cat#: 117307, clone: N418), PerCP/Cy5.5-conjugated anti-Cd11b (cat#: 101228, clone: M1/70), PE/Cy7-conjugated anti-Cd45 (cat#: 103114, clone: 30–F11), allophycocyanin (APC)-conjugated anti-Cd11b (cat#: 101212, clone: M1/70) and APC-conjugated anti-Cx3cr1 (cat#: 149008, clone: SA011F11). For the RT-qPCR or immunoblotting experiments of Cd11b$^+$/Cx3cr1$^+$ or Cd11b$^+$ cells,

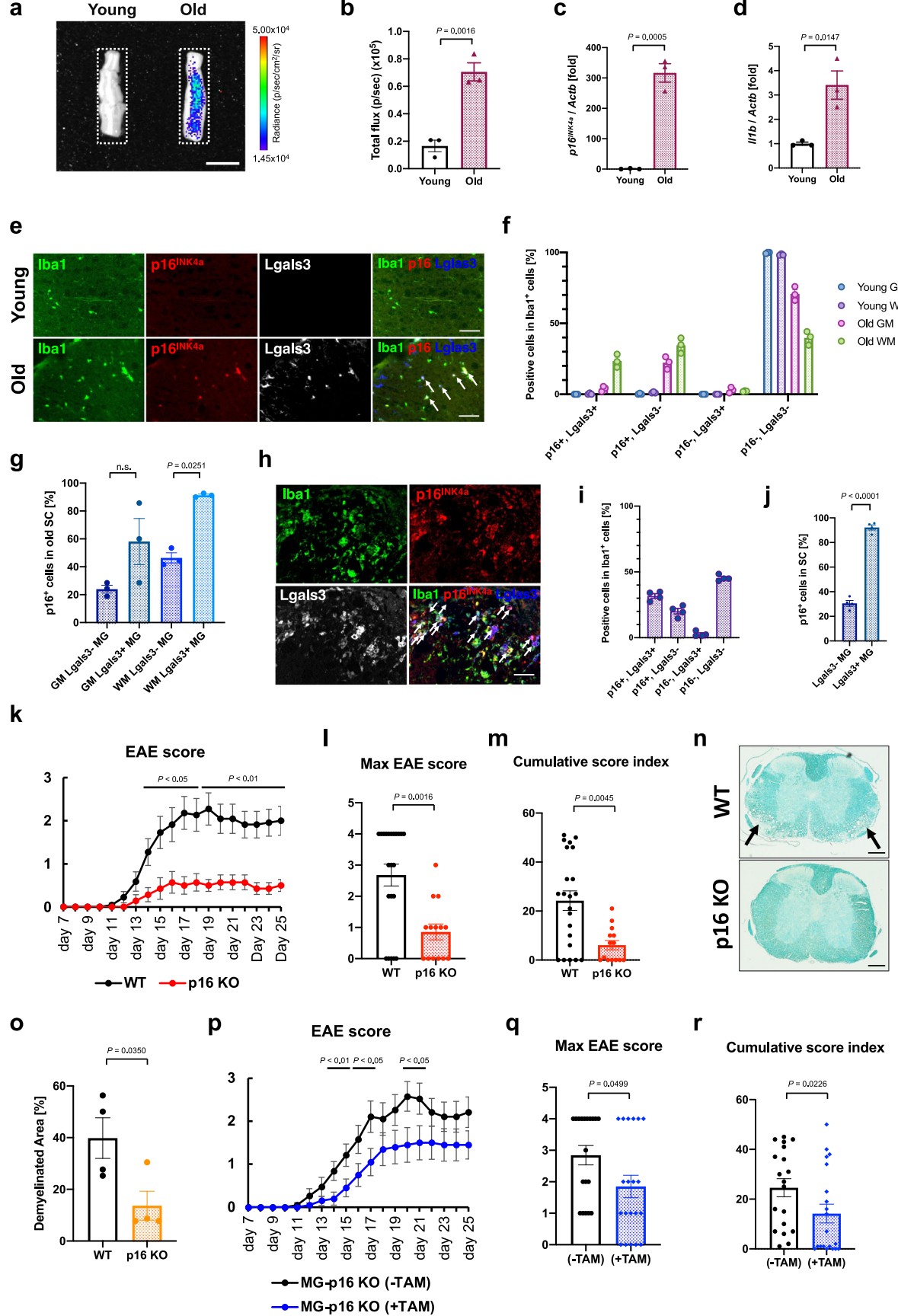

**Fig. 5 p16^INK4a knockout suppresses spinal cord demyelination induced by EAE. a** The spinal cords of female *p16-luc* mice (young: 2–3 months, old: 23–28 months) were isolated and subjected to bioluminescence imaging. Representative images of the spinal cord are shown. Colour bar indicates minimum and maximum pixel thresholds used. **b** Bioluminescence intensity emitted from young and old mouse spinal cord, given as total flux measured in the indicated region from (**a**). **c, d** RT-qPCR analysis of *p16^INK4a* (**c**), and *Il1b* (**d**) mRNA in spinal cord lysates from young (2 months) or old (26 months) male mice. *Actb* was used as an internal control. **e** Representative immunofluorescence images for Iba1 (green), p16^INK4a (red) and Lgals3 (white or blue) in the spinal cord white matter of young (2 months) or old (33 months) male mice. Arrows indicate triple-positive (Iba1$^+$/p16$^+$/Lgals3$^+$) cells.
**f** Quantification of the percentages of p16$^+$/Lgals3$^+$, p16$^+$/Lgals3$^-$, p16$^-$/Lgals3$^+$ and p16$^-$/Lgals3$^-$ Iba1$^+$ cells in the grey matter (GM) or white matter (WM) of spinal cord (SC) in young or old mice (young: $N = 3$, $n = 55$–121 (in GM), $n = 102$–112 (in WM) Iba1$^+$ cells; old: $N = 3$, $n = 101$–136 (in GM), $n = 156$–180 (in WM) Iba1$^+$ cells per mouse). **g** Percentages of p16^INK4a positive Iba1$^+$/Lgals3$^-$ or Iba1$^+$/Lgals3$^+$ cells in the spinal cord of old mice, related to (**f**). **h** Representative immunofluorescence images of spinal cord from young (2 months) female mice in which EAE was induced. Mice were sacrificed at the peak of the acute phase of EAE (day 22). Arrows indicate triple positive (Iba1$^+$/p16$^+$/Lgals3$^+$) cells. **i** Quantification of the percentages of p16$^+$/Lgals3$^+$, p16$^+$/Lgals3$^-$, p16$^-$/Lgals3$^+$ and p16$^-$/Lgals3$^-$ Iba1$^+$ cells in (**h**) ($N = 4$, $n = 109$–129 Iba1$^+$ cells per mouse). **j** Quantification of p16^INK4a-positive Iba1$^+$/Lgals3$^-$ or Iba1$^+$/Lgals3$^+$ cells in (**i**). **k** EAE daily clinical scores from female WT ($N = 22$) and p16 KO ($N = 14$) mice. **l, m** Max EAE score (**l**) and cumulative EAE score (**m**) in (**k**). **n** Representative images of spinal cord sections stained with luxol fast blue. Arrows indicate the area of demyelination.
**o** Quantification of demyelinated area in the spinal cord white matter of female WT and p16 KO mice. **p** EAE daily clinical score from non-treated ($N = 19$) and tamoxifen-treated ($N = 20$) female Cx3cr1-CreER^T2: p16^flox/flox mice. **q, r** Max EAE score (**q**) and cumulative EAE score (**r**) in (**p**). Data presented as mean ± S.E.M from three mice (for **b, c, d, f, g** and **j**) or four mice (for **o**). Statistical significance was determined with two-tailed unpaired Student's *t*-test (**b, c, d, j** and **o**), Mann Whitney test (**k, l, m, p, q** and **r**). and one-way analysis of variance (ANOVA) followed by Tukey's multiple comparison test (**g**). n.s., non-significant. The exact *P*-values for each day are given in Supplementary Data 1 (for **k** and **p**). Scale bar, 5 mm (for **a**), 50 μm (for **e** and **h**) or 200 μm (for **n**). MG microglia.

---

Cd11b$^+$ cells were condensed using Anti-FITC MicroBeads (Miltenyi Biotec, cat#: 130-048-701) or Anti-APC MicroBeads (Miltenyi Biotec, cat#: 130-090-855) and MS Columns (Miltenyi Biotec, cat#: 130-042-201) or LS Columns (Miltenyi Biotec, cat#: 130-042-401) according to the manufacturer's protocol. Before analysis, cells were filtered through a 40 μm cell strainer and stained with propidium iodide (PI). PI-negative, Cx3cr1-positive and Cd11b-positive cells (young mice, 100,000–180,000 cells; old mice, 110,000–170,000 cells) were sorted into 1.5 ml Protein LoBind Tubes (Eppendorf, Germany, cat#: 003018116) containing 1% BSA in HBSS and then centrifuged. After removal of the supernatant, the remaining pellet was lysed and total RNA was isolated according to the standard protocol of the RNeasy Micro Kit (Qiagen, The Netherlands, cat#: 74004). For the analysis of immune cells in the spinal cords and spleens of EAE-induced mice, the spinal cords were isolated and treated according to the same protocol as above. The spleens were coarsely minced with microscope slides and the cells were incubated with Red Blood Cell Lysis Buffer (Biolegend, cat#: 420301) for 1 min, followed by staining with fluorescence-conjugated antibodies for 10 min. The following antibodies were used: Brilliant Violet 421™-conjugated anti-CD45R/B220 (cat#: 103240, clone: RA3–6B2), Brilliant Violet 421™-conjugated anti-Ly6g (cat# 127628, clone: 1A8), FITC-conjugated anti-Cd45 (cat#: 103108, clone: 30–F11), FITC-conjugated anti-Cd11b (cat#: 101206, clone: M1/70), PE-conjugated anti-Cd11c (cat#: 117307, clone: N418), PE-conjugated anti-Cd8a (cat#: 100708, clone: 53–6.7), PerCP/Cy5.5-conjucated anti-Cd11b (cat#: 101228, clone: M1/70), PerCP/Cy5.5-conjucated anti-Cd45 (cat#: 103132, clone: 30–F11), PE/Cy7-conjugated anti-F4/80 (cat#: 123114, clone: BM8), PE/Cy7-conjugated anti-Cd4 (cat# 100528, clone: RM4–5) and APC-conjugated anti-CD3ε (cat#: 100312, clone: 145–2C11). Data were collected with a SONY SH800 Sorter (SONY, Japan) and Attune Nxt Autosampler (ThermoFisher Scientific, MA, USA), and analysed with the FlowJo™ software v10.8 (BD Biosciences, OR, USA).

**Immunoblotting**. Mice were anaesthetised by intraperitoneal injection of three anaesthetic agents and heparin (Mochida Pharmaceutical, Japan, cat#: 224122557) and transcardially perfused with ice-cold Hanks' Balanced Salt Solution (HBSS) (Nakarai Tesque, Japan, cat#: 17461-05) containing heparin. Isolated brain tissues in RIPA buffer containing 1% Protease inhibitor cocktail (Nacalai Tesque) were homogenised with beads using a lysis and homogenization system (Precellys) at 6000 rpm for 20 sec. After determination of protein concentration using the Protein Quantification Assay (Takara Bio Inc., Japan), all samples were denatured in Laemmli sample buffer (BIO-RAD, CA, USA) for 5 min at 95 °C. For Cd11b$^+$ cells separated with MACS column (young mice, 100,000–150,000 cells; old mice, 60,000–100,000 cells), cells were lysed directly with Laemmli sample buffer and denatured. Denatured samples were separated by SDS-polyacrylamide gel electrophoresis and transferred onto a polyvinylidene difluoride membrane (Merck Millipore). After blocking with 5% BSA in TBST or Blocking One (Nacalai Tesque, cat#: 03953-95) for 60 min, the membranes were incubated with the primary antibodies as follows: β-actin (1:10000, Merck Millipore, cat#: A5316), p16^INK4a (1:2000, abcam, UK, cat#: ab211542) or γH2AX (1:1000, abcam, cat#: ab2893) in Can Get Signal Solution 1 (TOYOBO, Japan, cat#: NKB-101). Membranes were then incubated with the secondary antibodies (1:2000, Cell signaling Technology, MA, USA) in Can Get Signal Solution 2 (TOYOBO, cat#: NKB-101) and visualised with Amersham ECL prime or select (GE Healthcare, IL, USA), followed by detection with chemiluminescence using the LAS-3000mini imaging system (Fujifilm, Japan) and analysis of data using Fiji. Semi-quantification of p16^INK4a

protein levels was performed by measuring the band intensity of p16^INK4a and β-Actin with Fiji. Uncropped and unedited blot images are included as Supplementary Fig. 14.

**Immunostaining analysis**. Mice were transcardially perfused with ice-cold PBS under anaesthesia. Mouse brain tissues or lumbar spinal cords were postfixed with Bouin's solution (Muto Pure chemicals, Japan, cat#: 33142) for 12-16 h at 4 °C and followed by a paraffin burial technique. Bouin's solution was used as a fixative to amplify the nuclei signal of p16^INK4a and Iba1. Tissues were cut into 5 μm thick sections using a microtome, deparaffinised in PathoClean (Fujifilm) and rehydrated sections were treated with microwave-oven heating in 10 mM citrate buffer (pH 6.0) for antigen retrieval. Brain sections were washed with PBS containing 0.2% Triton X-100 for 30 min at room temperature (RT) and then incubated TNB buffer (0.1 mM Tris-Hcl (pH7.5), 0.15 M NaCl and 0.5% TSA Blocking Reagent (Perkin Elmer, MA, USA, cat#: FP1012)) with 5% normal donkey serum for 1 h at RT. The sections were incubated with primary antibodies at 4 °C overnight and then reacted with fluorescently labelled secondary antibodies for 1 h at RT. The following primary antibodies were used for immunofluorescence staining: rabbit anti-p16 (1:200, cat#: ab211542, abcam), rabbit anti-Iba1 (1:1000, Fujifilm, cat#: 019-19741), guinea pig anti-Iba1 (1:1000, Synaptic Systems, Germany, cat#: 234004) and rat anti-Lgals3 (1:3000, Biolegend, cat#: 125401). The following secondary antibodies were used: Alexa Fluor Plus 488-conjugated donkey anti-rabbit IgG (1:2000; ThermoFisher Scientific, cat# A32790), Alexa Fluor Plus 488-conjugated donkey anti-guinea pig IgG (1:2000; JacksonImmunoResearch, UK, cat#: 706-545-148), Alexa Fluor Plus 555-conjugated donkey anti-rabbit IgG (1:2000; ThermoFisher Scientific, cat#: A32794), Alexa Fluor Plus 647-conjugated donkey anti-rat IgG (1:2000; ThermoFisher Scientific, cat#: A48272). For staining the nuclei, 4,6-diamidino-2-phenylinodole (DAPI; 0.5 μg/mL, Merck Millipore, cat#: D9542) was added to the secondary antibody solution. For immunohistochemistry, samples were blocked using Streptavidin/Biotin Blocking Kit (VectaStain Elite ABC Reagent Peroxidase, Vector Laboratories, CA, USA, cat#: PK-6100) and stained with the primary antibody against p16 (1:200, abcam, cat#: ab211542). Signal was detected with ImmPACT DAB EqV Peroxidase Substrate (Vector Laboratories, cat# SK-4103-100). Cell nuclei were counterstained with hematoxylin (Dojindo, Japan). To avoid autofluorescence especially in old mouse brain, all samples were treated with 1:40 of TrueBlack® Lipofuscin Autofluorescence Quencher (Cosmo Bio, Japan, cat#: 23007) in 70% ethanol for 1 min at RT after incubation with 2nd antibodies. The slides were mounted by ProLong Glass Antifade Mountant (ThermoFisher Scientific, cat#: P36984) or PARAmount-N (FALMA, Japan, cat#: 308-400-1). For WT and SOD1 G93A mouse (JAX stock #004435) spinal cord staining provided by Prof. Dr. Koji Yamanaka, the lumbar spinal cord was postfixed with 4% paraformaldehyde for 12-16 hours at 4 °C and then incubated with 30% sucrose for 1 day. The tissue was embedded with Tissue-Tek OTC Compound (Sakura Fine tech, Japan, cat#: 4583) and frozen with liquid nitrogen, and then cut into 10 μm thick sections with a microtome and stained. Fluorescence and histological images were observed and photographed using all-in-one fluorescence microscope (BZ-X710 or BZ-X800; Keyence, Japan). To quantify the number of Iba1, p16, Lgals3, and/or EdU positive cells in each region (corpus callosum, medulla oblongata, or spinal cord), pictures covering the entire region were taken using a 20× lens in the stitching mode of the BZ-X Analyzer software (Keyence); stitched images were created using the BZ-X Analyzer software. Subsequently, the number of Iba1$^+$, p16$^+$, EdU$^+$, and/or Lgals3$^+$

cells in the corresponding region was measured using the BZ-X Analyzer Software or cell counter in Fiji. The number of cells analysed is indicated in the figure legends.

**Quantitative real-time PCR**. Quantitative real-time PCR was performed by the following procedure[38]. Briefly, total RNA was extracted using RNeasy mini or micro kit (Qiagen) according to the manufacturer's protocol and cDNA was synthesised using a PrimeScript RT reagent kit with gDNA Eraser (Takara Bio Inc., Japan, cat.#: RR047B) or PrimeScript RT reagent kit (Takara Bio Inc., cat.#: RR037B). Quantitative real-time RT-PCR was performed on a Thermal Cycler Dice Real Time System III TP970 (Takara Bio Inc.) using TB Green® Premix Ex Taq™ (Tli RNaseH Plus) (Takara Bio Inc., cat#: RR820B). The mRNA expression levels of each gene were calculated relative to *Actb* expression levels. The PCR primer sequences used in this paper are as follows: *Actb* 5'-GATGACCCAGATCATGTTTGA-3' (forward) and 5'-GGAGAGCATAGCCCTCGTAG-3' (reverse); *p16^INK4a* 5'-GAACTCTTTCGGTC GTACCC-3' (forward) and 5'-CGAATCTGCACCGTAGTTGA-3' (reverse); *Cd11c* 5'-CTGGATAGCCTTTCTTCTGCTG-3' (forward) and 5'- GCACACTGTGTCC GAACTCA-3' (reverse); *Apoe* 5'- CTGACAGGATGCCTAGCCG-3' (forward) and 5'-CGCAGGTAATCCCAGAAGC-3' (reverse); *Axl* 5'-ATGGCCGACATTGCC AGTG-3' (forward) and 5'- CGGTAGTAATCCCCGTTGTAGA-3' (reverse); *Il1b* 5'-TGACGGACCCCAAAAGATGAAGG-3' (forward) and 5'- CCACGGGAAAGA-CACAGGTAGC-3' (reverse); *Cxcl10* 5'- CCAAGTGCTGCCGTCATTTTC-3' (forward) and 5'- GGCTCGCAGGGATGATTTCAA-3' (reverse).

**Bioluminescence imaging and image acquisition**. All imaging experiments were performed using an IVIS Lumina XRMS Series III (Perkin Elmer, MA, USA). For the detection of luminescence, mice were anaesthetised by intraperitoneal injection of the three anaesthetic agents. Mice were then injected intraperitoneally with 100 µl d-luciferin substrate (30 mg/ml in PBS, Fujifilm, cat#: 128-06911). The brains and the spinal cords were isolated 5 min after substrate injection, and the dorsal side of the brains were imaged for 5 min, followed by imaging of the ventral side of the brains for 5 min and the spinal cord for 5 min. Bioluminescence imaging data were analysed with Living Image Software 4.7.3 (Perkin Elmer).

**EAE induction**. Female mice (7–10 weeks old) were prepared for acclimatisation at least 1 week before the experiment and immunised with an emulsion containing 150 µg of I-A^b MOG_{35-55} peptide (MEVGWYRSPFSRVVHLYRNGK; MBL Life science, Japan, cat#: TS-704-P) and 500 µg of Mycobacterium tuberculosis extract H37RA in complete Freund's adjuvant (Chondrex, WA, USA, cat#: 7023) by subcutaneous injection on the both sides of the back. 400 µg of Pertussis toxin (List Biological Laboratories, CA, USA, cat#: 180) diluted in D-PBS (nakarai tesque, cat#: 14249-24) was intraperitoneally administrated at 0- and 48-h post-immunization. EAE scores were assessed daily according to the following criteria: 0, no abnormalities; 1, loss of tail reflex; 2, limp tail and hind limb weakness or abnormal gait; 3, paralysis of one hind-limb; 4, paralysis of two hind-limbs.

**Luxol Fast Blue staining**. The tissue sections on the slide were dehydrated with ethanol at various concentrations and PathoClean, and then embedded in paraffin. Myelin was detected using Luxol Fast Blue Stain Kit (Scy tec Laboratories, USA, cat#: LBC-1). The sections were incubated in Luxol Fast Blue Solution for 2 h at 60 °C. After rinsing sides in distilled water, sections were differentiated by dipping in 0.05% Lithium Carbonate solution for 20 s and then washed in 70% for 2 min. The percentage of demyelinated area to total white matter area of the spinal cord in each image was quantified using Fiji.

**in vivo proliferation assay**. Young (2-month-old) or old (28-month-old) WT mice were intraperitoneally injected with 5-ethynyl-2'-deoxyuridine (EdU) (50 mg/kg, 5 mg/ml in saline, Fujifilm, Japan, cat#: 052-08843) seven times in total for 2 weeks (on d1, d3, d5, d7, d9, d11, and d13) and sacrificed 2 days after the last injection (on d15). For old mice treated with PLX3397 (Selleckchem, TX, USA, cat#: S7818) for 2 weeks, the mice were intraperitoneally injected with EdU daily for 8 days (on d16, d17, d18, d19, d20, d21, d22, and d23) and sacrificed 6 days after the last injection (on d29). EdU incorporation was detected using a Click-iT Plus EdU Alexa Fluor 647 imaging kit (ThermoFischer Scientific, cat#: C10640).

**Single-cell RNA-sequencing of mouse brains**. Mice (a young and an old C57Bl/6 J mice for Figs. 2, 3, and Supplementary Figs. 2, 5, 13, and a young C57Bl/6 N GF, a B6 albino p16-luc old GF, young SPF and old SPF mice for Fig. 4 and Supplementary Figs. 7, 13) were anaesthetised by intraperitoneal injection of three anaesthetic agents and heparin and transcardially perfused with ice-cold HBSS containing heparin. The corpus callosum (for Figs. 2, 3) or medulla oblongata (for Fig. 4) of each mouse brain was isolated and roughly minced by scissors, followed by digestion according to the standard protocol of Neural tissue dissociation kits (P) (Miltenyi Biotec, cat#: 130-092-628). The brain homogenate was filtered through a 70 µm cell strainer, and cell debris was removed by centrifugation according to the standard protocol of Debris Removal Solution. The resulting solution including the cells was then filtered through a 40 µm cell strainer and counted using a hemocytometer. Single cell suspensions were processed using the 10x Genomics Chromium Controller according to the

protocol described in the Chromium Single Cell 3' Reagent Kits User Guide. During the process, Chromium Next GEM Single Cell 3' GEM, Library & Gel Bead Kit v3.1 (cat#: PN-1000128, 10x Genomics, USA), Single Index Kit T Set A (cat#: PN-1000213) and Chromium Next GEM Chip G Single Cell Kit (cat#: PN-1000127) were used. Approximately 16,500 live cells per sample were loaded onto the Chromium controller according to the manufacturer's protocol to generate 10,000 emulsions containing single cells and gel beads to prepare library and perform sequencing. Oil droplets containing encapsulated single cells and barcoded beads (GEMs) were then reverse-transcribed in a Veriti Thermal Cycler (ThermoFisher Scientific). Resulting cDNA labelled with a unique molecular index (UMI) and a cell barcode was then amplified to generate single cell libraries following the protocol outlined in the manufacturer's protocol. An Agilent Bioanalyzer High Sensitivity DNA Assay (Agilent High-Sensitivity DNA Kit, Agilent, CA, USA, cat#: 5067-4626) was utilised for the quantification. The amplified cDNA was then enzymatically fragmented, end-repaired and labelled with polyA. Illumina sequencing adaptors were ligated to the size-selected fragments and subsequently purification and size selection of the amplified cDNA was performed using SPRIselect magnetic beads (SPRIselect, Beckman-Coulter, CA, USA, cat#: B23317). Sample indices were finally selected and amplified, and then double-sided size selection was performed using SPRIselect magnetic beads. The quality of the resulting library quality was assessed using an Agilent Bioanalyzer High Sensitivity DNA assay.

**Analysis of mouse single-cell RNA-sequencing data**. To create scRNA-seq libraries, samples were sequenced on an Illumina NovaSeq 6000 in paired-end mode (read1: 28 bp; read2: 91 bp) at the RIMD. The raw sequencing reads were aligned using the Cell Ranger (v4.0.0) analysis pipeline (10x Genomics) to the mouse reference genome mm10, and unique molecular identifier (UMI) count matrices were generated. The generated matrices were input to a downstream analysis using Seurat (v3.2.3)[62]. Cells that contained >10% of the mitochondrial genes, genes observed in only <3 cells, and cells with <200 expressed genes were excluded from the analysis. To remove suspected doublets, DoubletFinder (v2.0.3)[63] was applied and detected doublets were excluded from the analysis. UMI counts were normalised and scaled for each cell. Canonical correlation analysis (CCA) was performed for batch corrections between samples with different sequence runs. For dimensionality reduction, principal component analysis (PCA) was performed using the top 2000 highly variable genes in the dataset. Clusters were then defined using the first to thirtieth principal components and shared nearest neighbour (SNN) graphs, and visualised using uniform manifold approximation and projection (UMAP). Marker gene detection and differentially expressed gene analysis of each cell cluster were performed using MAST (v1.14.0)[64] and adjusted *P*-value is based on bonferroni correction. Inference of gene regulatory networks was performed using the SCENIC pipeline[65]. Clusters positive for either Spi1 or Aif1 were extracted and the microglial population expressing microglia-specific genes and not other cell type-specific genes (e.g. Pf4 in macrophages) was used for subsequent microglial reclustering. Abbreviations: oligodendrocyte precursor cells (OPC), oligodendrocytes (OLG), astrocytes (ASC), ependymocytes (EPC), neuronal-restricted precursors (NRP), immature neurons (ImmN), mature neurons (mNEU), choroid plexus epithelial cells (CPC), endothelial cells (EC), pericytes (PC), vascular smooth muscle cells (VSMC), haemoglobin-expressing vascular cells (Hb-VC), vascular and leptomeningeal cells (VLMC), microglia (MG), monocytes (MNC), macrophages (MAC), T cells (T), neutrophils (NEUT), dendritic cells (DC) and B cells (B). The raw data of the scRNA-seq analysis were deposited in the DNA Data Bank of Japan (DDBJ) (DRA014178).

**Bulk RNA sequencing of microglia in young and old GF and SPF mice**. Young and old GF and SPF mice (albino p16-luc background, young GF, 2 months, N = 2; young SPF, 2 months, N = 2; old GF, 22 months, N = 2; old SPF, 22 months) were anaesthetised by intraperitoneal injection of three anaesthetic agents and heparin and transcardially perfused with ice-cold HBSS containing heparin. The brains were cut in sagittal midlines and the cerebral hemispheres were roughly minced by scissors, followed by digestion according to the standard protocol of Neural tissue dissociation kits (P). The brain homogenate was filtered through a 70 µm cell strainer, and cell debris was removed by centrifugation according to the standard protocol of Debris Removal Solution. The cells in HBSS with 1% BSA were then treated on ice with CD16/32 FcR-blocking reagent for 10 min, followed by 15 min of staining with fluorescence-conjugated antibodies. The following antibodies were used (all antibodies were purchased from BioLegend): FITC-conjugated anti-Cd45 (cat#: 103108; clone, 30F–11) and APC-conjugated anti-Cd11b (cat#: 101212; clone, M1/70). Before sorting, cells were filtered through a 40 µm cell strainer and stained with propidium iodide (PI). PI-negative, Cd45-low, and Cd11b-positive cells (young SPF, 100,000 cells; young GF, 150,000 cells; old SPF, 50,000 cells; and old GF 50,000 cells) were sorted into 1.5 ml Protein LoBind Tubes containing 1% BSA in HBSS and then centrifuged. After removal of the supernatant, the remaining pellet was lysed and total RNA was isolated according to the standard protocol of the RNeasy Micro Kit. Full-length cDNA was prepared using a SMART-Seq HT Kit (Takara Bio, Japan) according to the manufacturer's instructions. According to the SMARTer kit instructions, an Illumina library was then prepared using a NexteraXT DNA Library Preparation Kit (Illumina). An Illumina NovaSeq 6000 sequencer (Illumina) were used for DNA Sequencing in

100-base paired-end mode. Adaptor trimming was conducted by Trimmomatic and sequenced reads were then mapped to mouse reference genome sequences (mm10) by using TopHat (v2.1.1). Cuffnorm (v2.2.1) was used for calculating Fragments per kilobase of exons per million mapped fragments (FPKM). The raw data in the RNA-seq analysis were deposited in the Gene Expression Omnibus database (GEO, GSE204828). iDEP.95 was used for generating heatmap, a figure of PCA analysis and gene ontology pathway tree.

**Statistics and reproducibility**. Statistical significance was determined with two-tailed unpaired Student's $t$-test (Figs. 1b, 2f, 2i, 2k, 2l, 3i, 3k, 3l, 4a, 4k, 4m, 4n, 4o, 5b, 5c, 5d, 5j, 5o, Supplementary Figs. 1f, 3c, 4c, 4d, 4g, 9a, 9c, 9d, 9e, 10b, 10c, 10d, 10e, 10f, 10g, 10h, 10i, 10j, 10k, and 11e), Mann Whitney test (Figs. 5k, 5l, 5m, 5p, 5q, 5r) or one-way analysis of variance (ANOVA) followed by Tukey's multiple comparison test (Figs. 4p, 5g). $P$-values < 0.05 were considered significant (n.s., non-significant). All experiments were repeated at least twice independently with similar results.

**Reporting summary**. Further information on research design is available in the Nature Portfolio Reporting Summary linked to this article.

## Data availability

The raw data of scRNA-seq analysis were deposited in the DDBJ (DRA014178) and the raw data in the RNA-seq analysis were deposited in the GEO (GSE204828). The source data for the figures in this study are provided in Supplementary Data 1. All other data are available from the corresponding author.

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

## Acknowledgements

We acknowledge the NGS core facility of the Genome Information Research Center at the RIMD (Osaka University) for the support in RNA sequencing and data analysis. We also thank Ms. Masae Suzuki and Ms. Yuko Wakabayashi for caring for the experimental mice and other members in our laboratory for discussion during the preparation of this manuscript (Osaka University). We thank Dr Anaelle Dumas for her help in creating the figures using elements from BioRender (Fig. 2a, Supplementary Figs. 4a, 4e, 11c and 12). This work was supported in part by Japan Agency of Medical Research and Development (AMED) (grant JP21gm5010001, JP22gm1710004 and JP22zf0127008), Japan Society for the Promotion of Science (JSPS) (grant JP22H00457) and the Japan Science and Technology Agency (grant JPMJMS2022-15) to E.H., and JSPS (20J01264) to T.Matsudaira. This work was also supported in part by grants from AMED (grants JP21wm0425010 and JP21gm1510006), JSPS (grants JP21H05694 and JP22H02944) to T.H. and the Collaborative Research Program of Institute for Protein Research, Osaka University, ICR-21-3.

## Author contributions

Ta.M. designed experiments, performed most of the experiments, analysed the data and wrote the manuscript. S.N. helped the experiments related to EAE experiments and GF mice. Y.K. analysed the scRNA-seq data. S.K. and K.U. established the p16 staining system and helped with the experiment using *p16-luc* mice and GF mice. T.K. helped with the sample collection and experiments related to immunofluorescence. K.S., T.O., T.H., Y.F. and T.Y. analysed the data. O.K. and K.Y. provided the samples from SOD1 G93A mice and analysed the data. To.M. analysed the data and helped write the manuscript. E.H. helped write the manuscript and supervised the project.

## Competing interests

The authors declare no competing interests.
