## [Peer Review File · Communications Biology]

Cellular senescence in white matter microglia is induced during ageing in mice and exacerbates the neuroinflammatory phenotypeReviewers' comments:

Reviewer #1 (Remarks to the Author):

In this article, Matsudaira et al study the presence of senescent microglia in the pathological brain. This study generally recapitulates findings that others have reported before (see my specific comment) and expands those to evidence the presence of microglial senescence in the context of modification of the gut microbiome as well as in a model of EAE. The experiments are generally well conducted, and well connected, helping to inform about the different aspects characterising a senescent cell. Overall, this article contains a number of valuable findings, worth reporting to the broader field. However, there needs to be a more open, transparent, acknowledgement of the previous research in this area, in order to better frame the obtained results.

Main comments:

1. It kind of surprising how shallow and outdated the literature review is, and how key papers have been forgotten, or not recognised as prior knowledge in the introductory section. These include seminal, critical, papers by the Baker group (senescence in microglia in tau pathology), Spillantini group (senescence in microglia in tau pathology), and Gomez-Nicola group (senescence of microglia in Abeta pathology). The latter is particularly central to the current submission, since it explored for the first time the linkage of senescent microglia with DAMs, which somehow is presented by the current submission as a novel idea, but really it is not. It is important to acknowledge research by others, as this builds up and increases the progress of science, as well as helps identifying what is novel and what is not.
2. Linked to the previous comment, the results presented here are mostly not novel, albeit useful as a confirmation of previously published studies (using other reporters). For example, p16 reporter mice have been used by many others to detect senescent cells in the brain, and this should be acknowledged more openly. However, I have to say that the experiments reported here are well connected and serve as very strong validation of previous findings, and in my opinion this doesn't affect the value of the study. The data using EAE model, as well as the gut microbiome intervention is indeed novel and useful.
3. In relation to the p16-luc mice, it's not clear how sensitive this model is, in terms of correlating p16-luc signal with actual gene expression. Experiments supporting the ability of p16-luc to detect p16 expression in a sensitive and correlative fashion would be useful to understand if the detected cells are a true reflection of the underlying expression.
4. The single-cell experiment shown in Figure 2 is analysed in an unconventional way, splitting young and old and not showing initial combined clusters and how represented these are by age. The enrichment of Cdkn2a is powerful, but somehow less informative than using a more complex signature of senescence, as for example done in Hu et al., 2021.

Reviewer #2 (Remarks to the Author):

In this study Matsudaira and colleagues show that microglia are in a state of senescence in the aged central nervous system, particularly in the aged corpus callosum and spinal cord in mice. The senescent microglia population overlaps to some extent with the previously described disease associated microglia (DAM) phenotype. They further show that the presence of commensal gut microbiota promotes the development of both the senescent and the DAM microglia phenotype in the aging brain and that systemic and cell autonomous knock out of p16INK4a can inhibit the emergence of these microglia states. The authors used a wide array of techniques and mouse models (p16-luc,) to explore the distribution and role of p16 positive senescent microglia in the aging brain and under neuroinflammatory conditions, which is a strength of this study. However, the finding that p16 positive cells accumulate in the aged brain and that they mainly correspond to microglia is not novel (PMID:

34598318). Neither is the observation that the depletion of p16 positive cells has beneficial effect in mouse models of neurodegeneration (PMID: 30232451, PMID: 22048312), nor is the finding that the composition of gut microbiota contributes to microglia aging (reviewed in PMID: 35349746). Nonetheless, there is enough novelty in the present study that would warrant publication in Communications Biology – the detailed dissection of the relationship between p16+ microglia and the DAM phenotype is novel and could open up a potentially interesting new area of research. Accordingly, it is suggested that the authors (using their already existing data/figures) restructure the paper in order to highlight their novel findings and conclusions, rather than spending a large part of their paper to provide further experimental support to findings that have already been published, and are distracting from the novel information presented in this study.

Separately, there are a couple of major and minor points which should be addressed.

Major points:

- the authors failed to discuss that in vivo EdU incorporation assay can also capture DNA damage repair not only proliferation, and that the incorporated EdU can also cause DNA damage and has cytotoxic properties – these characteristics of the applied assay are important for how the resulting data is interpreted
- the authors failed to show based on what they called microglia either CD11c- or CD11c+ (Figure) – ideally isotype control staining should have been used to determine the threshold – but even if there was an isotype staining control, the distribution of CD11c staining (Supplementary figure 5) on microglia does not suggest that there is a positive and a negative population – the authors should rename these CD11chigh and CD11clow populations – the current nomenclature is misleading and is not in line with the presented data
- it has been shown that enzymatic tissue digestion approach (used in this study) negatively affects the quality of the microglia single cell RNA-sequencing studies (PMID: 35260865) – the authors should discuss this and investigate where the reported stress signature microglia cluster fall in their study
- it would be important to provide better quality photomicrographs of immunofluorescence studies, with particular emphasis on the Iba1 staining, which should label microglial processes even in thin sections – most of the representative images shown for Iba1 staining in Figure 2g, Figure 2i, Figure 3i, Figure 4k, Figure 5e, Supplementary figure 3b, Supplementary figure 3f and Supplementary figure 8b do not look like Iba1 staining
- it would be really interesting to see how the DAM phenotype looks like in the Cx3cr1-CreERT2: p16flox/flox mice in aging/EAE, etc

Minor points:

- title – in its current form the title is not informative as to the major findings of the current study, we suggest the authors rephrase the title
- microglia do not reproduce – line 127: “induced transient depletion and subsequent microglia reproduction”
- line 462, line 693 – please provide the number of microglia cells used for the bulk RNA-sequencing studies
- line 492 – please provide the number of sorted microglia cells that were used for the Western blot analysis
- line 608 – it is not clear what was the EdU injection schedule, please rephrase/specify
- line 670 – please include these abbreviations in the legend of the relevant figures –figures and their legends should be self-explanatory

Reviewer #3 (Remarks to the Author):

In the manuscript entitled “Comprehensive analysis of the central nervous system revealed microglial senescence and its disease associations” by Matsudaira et al, the authors applied multiple analytical

technologies including state-of-the-art single-cell RNA-seq to assess the characteristics of senescent cells in the CNS. They defined microglia (especially DAM) as majority of senescent cells in the CNS white matter. Suppression of microglial senescence by genetic manipulation revealed attenuation of EAE severity. The study is overall very interesting and will provide important new information to the field. However, I have a number of concerns/suggestions that should be addressed/responded to improve the manuscript.

Concerns/suggested:

1. For scRNA-seq experiment, it is not clear how many mice were used (should be addressed in Figure legend). The results (e.g. Fig. 2D, 3B, 3F, 3G, etc.) should be shown for each mouse in each group. To evaluate and interpret the data it is important to know the biological variation of each feature. In this manuscript, the authors show the results of scRNA-seq only per cell or per group. This makes it impossible for the reader to evaluate the consistency and/or variation of each feature across different animals.
2. It's written on Line 410-411: The p16-luc mice and the EAE-induced mice were female, and the other mice were male. What does "the other" refer to? Why were different sexes used in different experiment? Sex of mice used in each experiment should be addressed in the figure legends.
3. It's written on Line 415-417: 100 ul of 20 mg/ml Tamoxifen dissolved in corn oil were injected intraperitoneally for five days in a row in Cx3cr1-CreERT2: p16flox/flox mice and then maintained 3-4 weeks before EAE experiments to reduce the effect of p16INK4 depleted lymphocytes such as monocytes and peripheral macrophages⁴⁷ What was the injection schedule of tamoxifen (daily, every two days,...?).
4. FACS antibody clones should be addressed.
5. It's written on Line 96-98: Further evaluation revealed that p16INK4a mRNA and protein were highly expressed in the brain's white matter regions, such as the corpus callosum and medulla oblongata in old mice (Fig. 1c, d). To increase readability, the authors should described in the text which further evaluation they used.
6. The IHC results shown in Figure 1e should be quantified.
7. Suppl. Figure 2: cluster-ID should be addressed along with labelling of cell type. The authors may directly label cell type on UMAP. And as mentioned in "1" the results shown in Suppl. Figure 2B should also be visualized per animal, not only per group or per cell.
8. In Figure 2, it seems that microglia is the most abundant cell type in the UMAP plot, thus showing only the number of cell expressing Cdkn2a may be misled. The proportion of Cdkn2a-positive cells (e.g. % of total microglia) should also be addressed.
9. Figure 2f: how many mice were used, (semi) quantitative value (per mouse) should be shown.
10. In some experiments, 33 months old mice were used instead of 22 or 24 months old. This is quite a big difference. Is there any reason of choosing these two different ages for the group of old mice? Would this have any impact on the results?
11. In Figure 2h-k and other similar figures, it was described "young: N = 3, n > 80 cells, old: N = 3, n > 200 cells per mouse". It would be much better to describe as a range. Also the authors should address how many cells were analyzed (thus e.g. n = x - y out of z cells analysed).
12. On Line 547, it was written "6x8 images for medulla oblongata". What does "6x8 images" mean? How many cells were analyzed in total?
13. Are there any changes in morphology of aged and/or senescent microglia, compared with young microglia? The authors should also show Iba-1-positive (macrophage/microglia) cells with high magnification to show these possible changes. Also, GF microglia were known to have different morphology compared to SPF microglia. This is impossible to evaluate with low-magnification images (e.g. Figure 4k).
14. Suppl. Figure 10: another method should be used to validate the model (i.e. expression of p16INK4a), for qPCR or IHC.
15. The differences between results shown in Figure 5K and 5P are quite interesting, the authors should discuss more about this, e.g. involvement of other cell types, insufficiency of know out, etc.
16. Suppl. Figure 9B-G: proportion (e.g. % of CD45+) should be shown along with absolute cell number.
17. At the end, the link between microglial senescence in aged mice and those in pathology that occur

in young mice is unclear. Also, although in young mice the microglial senescence is hardly detected, suppression of p16INK4a in young mice could provide significant effects in EAE. This needs some more discussion. Would this possibly mean that the disease causes microglial senescence and not another way around, but how suppression of this cascade can give positive effects on disease severity?

18. In the Methods section, an irradiation experiment was described. This experiment seems to be missing or unclearly described.

Point-by-point responses to the reviewers' comments

We sincerely thank both the reviewers and the editor for their constructive and thoughtful comments on how to improve our manuscript. We are grateful for their shared appreciation of our manuscript, such as **“Overall, this article contains a number of valuable findings, worth reporting to the broader field. (Reviewer 1)”**, **“Nonetheless, there is enough novelty in the present study that would warrant publication in Communications Biology. (Reviewer 2)”** and **“The study is overall very interesting and will provide important new information to the field. (Reviewer 3)”**.

All of the comments are quite helpful in improving our manuscript. We believe that by addressing the reviewers' concerns, we have produced a more robust and interesting manuscript. Our point-by-point responses to the reviewers' comments are detailed below, with the original comments in bold.

Reviewer #1 (Remarks to the Author):

In this article, Matsudaira et al study the presence of senescent microglia in the pathological brain. This study generally recapitulates findings that others have reported before (see my specific comment) and expands those to evidence the presence of microglial senescence in the context of modification of the gut microbiome as well as in a model of EAE. The experiments are generally well conducted, and well connected, helping to inform about the different aspects characterising a senescent cell. Overall, this article contains a number of valuable findings, worth reporting to the broader field. However, there needs to be a more open, transparent, acknowledgement of the previous research in this area, in order to better frame the obtained results.

Main comments:

1. It kind of surprising how shallow and outdated the literature review is, and how key papers have been forgotten, or not recognised as prior knowledge in the introductory section. These include seminal, critical, papers by the Baker group (senescence in microglia in tau pathology), Spillantini group (senescence in microglia in tau pathology), and Gomez-Nicola group (senescence of microglia in Abeta pathology). The latter is particularly central to the current

submission, since it explored for the first time the linkage of senescent microglia with DAMs, which somehow is presented by the current submission as a novel idea, but really it is not. It is important to acknowledge research by others, as this builds up and increases the progress of science, as well as helps identifying what is novel and what is not.

We would like to thank the reviewer for the feedback on our limited literature review. It is a totally plausible opinion. **We have revised the manuscript and highlighted our new sections in yellow.**

To answer your comment, we have added the necessary information to the Introduction by citing references cited by the reviewer and other important papers relevant to this research.

2. Linked to the previous comment, the results presented here are mostly not novel, albeit useful as a confirmation of previously published studies (using other reporters). For example, p16 reporter mice have been used by many others to detect senescent cells in the brain, and this should be acknowledged more openly. However, I have to say that the experiments reported here are well connected and serve as very strong validation of previous findings, and in my opinion this doesn't affect the value of the study. The data using EAE model, as well as the gut microbiome intervention is indeed novel and useful.

We are very grateful for your suggestion and explanation. To the best of our knowledge, p16-luc (Yamakoshi et al. 2009, PMID: 25923845), p16-3MR (Demaria et al. 2014, PMID: 25499914), and p16-CreERT2 R26-tdTomato (Omori et al. 2020, PMID: 32949498) mice were used as p16 reporter mice. The p16-3MR mice have been recently demonstrated to show senescent cells in the brain (Talma et al. 2021, PMID: 34598318), and we have cited the relevant papers in Introduction. Moreover, we mentioned p16-ATTAC mice, which can remove p16-expressing cells in vivo (Baker et al. 2011, PMID: 22048312; Baker et al. 2016, PMID: 26840489; Bussian et al. 2018, PMID: 30232451), in the Introduction to acknowledge previous useful mouse lines for detecting and/or removing senescent cells. Although detecting p16-expressing cells in the brains of old mice has been established, as mentioned by the reviewer, our research provides an interesting viewpoint, where senescent microglia may be linked to the phenotype of EAE, and the gut microbiota may contribute to the accumulation of senescent cells.

3. In relation to the p16-luc mice, it's not clear how sensitive this model is, in terms of correlating p16-luc signal with actual gene expression. Experiments supporting the ability of p16-luc to detect p16 expression in a sensitive and correlative fashion would be useful to understand if the detected cells are a true reflection of the underlying expression.

We sincerely appreciate the reviewer's important suggestion. The p16-luc mouse line comprises transgenic mice that carry the human p16^{INK4a} gene locus tagged with a firefly luciferase. To validate this mouse line, we have previously demonstrated the correlation between the level of human p16 expression, mouse endogenous p16 expression, and luciferase signal in the mouse body (Yamakoshi et al. 2009, PMID: 25923845). In these mice, the levels of human p16 expression, mouse endogenous p16 expression, and luciferase signalling are well correlated in the cervical LNs, lungs, mesenteric nerves, and testes (Yamakoshi et al. 2009, Figs. 1D–F).

Although our p16-luc mice efficiently represent endogenous p16 expression, in our previous study, we were unable to detect the luciferase signal in the brain because the signal was critically weakened by their anatomical depth. In this work, we focused on brain ageing and were able to detect the luciferase signal for the first time by observing the brains isolated from aged mice. Unfortunately, we do not currently have aged p16-luc mice, and it is difficult to quantitatively demonstrate the sensitivity of p16-luc mice to detect endogenous p16 expression in the CNS. However, at least the strong luciferase signal in the medulla oblongata was well correlated with the level of p16 expression in our data (Figs. 1a, 1c, 1d, and Supplementary Fig. 1a), suggesting that a strong luciferase signal is indicative of an accumulation of senescent cells in the CNS.

Figs. 1a, 1c, 1d

Supplementary Fig. 1a

4. The single-cell experiment shown in Figure 2 is analysed in an unconventional way, splitting young and old and not showing initial combined clusters and how represented these are by age.

We thank you for these comments. Considering your comment and that of reviewer #3, we have revised the figures to demonstrate the data derived from the brains of both young and old mice (Fig. 2b, Supplementary Fig. 2a). We have also shown how these are represented in the first combined clusters by age (Fig. 2c).

Figs. 2b, 2c

Supplementary Fig. 2a

We have also applied the same change for microglia reclustering data (Figs. 3a, 3b) and in the analysis of young/old GF and SPF mice (Figs. 4b, 4c and Supplementary Figs. 7a–c).

Figs. 3a, 3b

Figs. 4b, 4c

Supplementary Figs. 7a–c

The enrichment of Cdkn2a is powerful, but somehow less informative than using a more complex signature of senescence, as for example done in Hu et al., 2021.

This is indeed a very important and valid point. Hu et al. examined Cdkn1a, Cdkn2d, Casp8, Il1b, Glb1, and Serpine1 as indicators for senescence signature. Therefore, we also checked these genes and created a bubble plot in the reclustered data of microglia (Supplementary Fig. 5d). We were able to confirm even more clearly that the percentage of cells expressing Serpine1, Glb1, Il1b and Cdkn1a is highest in cluster 8, where the expression of Cdkn2a and DAM genes is also the highest (Fig. 3d). These findings are similar to the result that the microglial cluster with the highest DAM score also has the highest senescence level in APP/PS1 mice (Hu et al. 2021, PMID: 34107254), suggesting that the clusters with the highest DAM content in microglia are the most senescent during both aging and disease. We have represented the data in Supplementary Fig. 5d and revised the Results section (lines 161–164).

Supplementary Fig. 5d

Reviewer #2 (Remarks to the Author):

In this study Matsudaira and colleagues show that microglia are in a state of senescence in the aged central nervous system, particularly in the aged corpus callosum and spinal cord in mice. The senescent microglia population overlaps to some extent with the previously described disease associated microglia (DAM) phenotype. They further show that the presence of commensal gut microbiota promotes the development of both the senescent and the DAM microglia phenotype in the aging brain and that systemic and cell autonomous knock out of p16INK4a can inhibit the emergence of these microglia states. The authors used a wide array of techniques and mouse models (p16-luc,) to explore the distribution and role of p16 positive senescent microglia in the aging brain and under neuroinflammatory conditions, which is a strength of this study. However, the finding that p16 positive cells accumulate in the aged brain and that they mainly correspond to microglia is not novel (PMID: 34598318(Talma et al.)). Neither is the observation that the depletion of p16 positive cells has beneficial effect in mouse models of neurodegeneration (PMID: 30232451(Bussian et al.), PMID: 22048312(Baker et al.)), nor is the finding that the composition of gut microbiota contributes to microglia aging (reviewed in PMID: 35349746(Zhou et al. 2022)). Nonetheless, there is enough novelty in the present study that would warrant publication in Communications Biology – the detailed dissection of the relationship between p16+ microglia and the DAM phenotype is novel and could open up a potentially interesting new area of research.

We are very grateful for this statement, which acknowledges the main points of our study. In the revised manuscript, we have thoroughly discussed previous studies in the Introduction and Discussion.

In this study, we have shown that senescent microglia contribute to the exacerbation of the neuroinflammatory phenotype and demonstrated that proliferation and state change in the absence of microbiota in old mice. We have also elucidated the detailed relationship between p16-expressing microglia and DAM phenotype. These results will help us further understand the mechanism and function of senescent microglia.

Accordingly, it is suggested that the authors (using their already existing data/figures) restructure the paper in order to highlight their novel findings and conclusions, rather than spending a large part of their paper to provide further experimental support to findings that have already been published, and are distracting from the novel information presented in this study.

Thank you for this valuable suggestion. We should have prioritised data that present novel information simultaneously with previous results obtained in similar conditions. Therefore, **we have added new sentence–highlighted in yellow–that demonstrate the relationship between our findings and previous results in the Results section to emphasise the novelty of our work.** We hope that this will increase the readability for readers and meet the reviewer's requirements.

Separately, there are a couple of major and minor points which should be addressed.

Major points:

- the authors failed to discuss that in vivo EdU incorporation assay can also capture DNA damage repair not only proliferation, and that the incorporated EdU can also cause DNA damage and has cytotoxic properties – these characteristics of the applied assay are important for how the resulting data is interpreted

We greatly appreciate these comments. We revised the text and added a citation to highlight the shortcomings of the EdU incorporation assay in the Results section (lines 140–141).

- the authors failed to show based on what they called microglia either CD11c- or CD11c+ (Figure) – ideally isotype control staining should have been used to determine the threshold – but even if there was an isotype staining control, the distribution of CD11c staining (Supplementary figure 5) on microglia does not suggest that there is a positive and a negative population – the authors should rename these CD11c^{high} and CD11c^{low} populations – the current nomenclature is misleading and is not in line with the presented data

Thank you very much for the suggestion. We decided the threshold of Cd11c-high and Cd11-low using the microglia in young mice. Following the reviewer's advice, we have added data on young mice in Supplementary Fig. 6.

Supplementary Fig. 6

- it has been shown that enzymatic tissue digestion approach (used in this study) negatively affects the quality of the microglia single cell RNA-sequencing studies (PMID: 35260865) – the authors should discuss this and investigate where the reported stress signature microglia cluster fall in their study

Thank you very much for bringing this to our attention. As the reviewer noted, Marsh et al. reported ex vivo “activated” microglia (exAM), which are detected in the scRNA-seq analysis of sorted microglia when the brain tissue is enzymatically digested without transcription and translation inhibitors (Marsh et al. 2022; PMID: 35260865). We have additionally analysed the genes that exAM highly express, including Fos, Jun, Hspa1a, Dusp1, Ccl3, Ccl4, and Zfp36 in our scRNA-seq analysis of microglia reclustering data (Supplementary Fig. 5e).

Supplementary Fig. 5e

In our data, immediate early genes (Fos, Jun) and stress-induced genes (Hspa1a and Dusp1) are strongly upregulated in clusters 1 and 3. However, immune-signalling genes (Ccl3 and Ccl4) are increased in clusters 2, 4, 5 and 7. Moreover, the gene expression of anti-inflammatory protein Zfp36, which is also highly expressed in exAM, is relatively high in clusters 3 and 7. Interestingly, none of the genes associated with exAM are highly expressed in DAM-related clusters 6 and 8, where microglia strongly express Cdkn2a. Thus, although some of the microglia in our study were affected by the enzymatic digestion in our protocol, the DAM clusters we mainly focus on are less relevant to exAM. We have added the data to Supplementary Fig. 5e and discussed it in lines 164–168.

- it would be important to provide better quality photomicrographs of immunofluorescence studies, with particular emphasis on the Iba1 staining, which should label microglial processes even in thin sections – most of the representative images shown for Iba1 staining in Figure 2g, Figure 2i, Figure 3i, Figure 4k, Figure 5e, Supplementary figure 3b, Supplementary figure 3f and Supplementary figure 8b do not look like Iba1 staining

We would like to apologise for the quality of the images of immunofluorescence studies. As indicated, microglial branches should be usually observed by Iba1 staining. Unfortunately, we had to stain for Iba1 using tissues fixed with Bouin's fixation as described in the Methods section, which well preserves nuclear antigens, including p16 proteins, but leads to weakened Iba1 signals in microglial branches. It is considered challenging to detect p16 expressions in mouse tissues by immunostaining due to the lack of good antibodies and established protocols, and our success in p16 staining in the brain of aged mice using Bouin's fixation is one of the key achievements of this study. Nevertheless, we found that it is very difficult to see the branching of the microglia in the immunostaining for Iba1 after Bouin's fixation.

Iba1 staining in the cortex and corpus callosum of the brain of young and old mice

Young (male, 2m), Old (male, 33 m), scale bar, 50 μ m.

Iba1⁺ cells are indicated by arrows. Guinea pig anti-Iba1 antibody (1:1000, Synaptic Systems, cat#: 234004, 234308), which is commonly used, was used for staining in this study.

However, we have adjusted the intensity of the Iba1 staining shown in previous Figs 2g, 2i, 3i, 4k, and 5e and Supplementary Figs 3b, 3f, and 8b to more clearly present microglial processes.

Fig. 2h (previous Fig. 2g)

Fig. 2j (previous Fig. 2i)

j

Fig. 3j (previous Fig. 3i)

j

Fig. 4l (previous Fig. 4k)

l

Fig. 5e

e

Supplementary Fig. 4b (previous Supplementary Fig. 3b)

b

Supplementary Fig. 4f (previous Supplementary Fig. 3f)

f

Supplementary Fig. 9b (previous Supplementary Fig. 8b)

- it would be really interesting to see how the DAM phenotype looks like in the *Cx3cr1-CreERT2: p16flox/flox* mice in aging/EAE, etc

We are grateful for this suggestion. To answer your comment, we have examined the number of Lgals3-positive microglia in two models: (1) aged *Cx3cr1-CreERT2 p16-flox/flox* mice and (2) mice fed with a cuprizone diet that has been shown to induce DAM by eliciting demyelination (Nugent et al. 2020; PMID: 31902528). In the aged mice administered with tamoxifen (1) the number of Lgals3-positive microglia slightly decreased compared to their control mice without tamoxifen administration, but the difference was not significant ($P = 0.0580$). In the cuprizone model (2), the number of Lgals3-positive microglia was comparable between mice with and without tamoxifen ($P = 0.7993$). We have also shown that there was no significant difference in the number of Cd11c-positive DAM between WT and p16 KO mice in which EAE was induced (Supplementary Fig. 10b). As the DAM phenotype in p16-deficient mice remains unclear, we plan to investigate the DAM phenotype in more detail in future research using mice deprived of p16^{INK4a} in microglia.

(1)

Cx3cr1-CreERT²; p16^{INK4a} flox/flox

Medulla oblongata

(- TAM) male, 22 months, N = 3, n = 308-360 Iba1⁺ cells per mouse
(+ TAM) male, 22-24 months, N = 3, n = 348-377, Iba1⁺ cells per mouse
Data presented as mean ± S.E.M.
n.s., non-significant, two-tailed unpaired Student's t-test.

(2)

Cx3cr1-CreERT²; p16^{INK4a} flox/flox

Corpus callosum

(- TAM) male, 3 months, N = 3, n = 269-307 Iba1⁺ cells per mouse
(+ TAM) male, 3 months, N = 3, n = 270-321, Iba1⁺ cells per mouse
Data presented as mean ± S.E.M.
n.s., non-significant, two-tailed unpaired Student's t-test.

Minor points:

- title – in its current form the title is not informative as to the major findings of the current study, we suggest the authors rephrase the title

We are very grateful for this comment. To clarify the major findings of this work, we have revised the title to "Cellular senescence in white matter microglia is induced during ageing and exacerbates the neuroinflammatory phenotype."

- microglia do not reproduce – line 127: "induced transient depletion and subsequent microglia reproduction"

Thank you very much for bringing this to our attention. We have revised this term to "repopulation."

- line 462, line 693 – please provide the number of microglia cells used for the bulk RNA-sequencing studies

Thank you for your comment. We have discussed the information on the number of sorted microglia in the Methods section (lines 383–384, 625–626).

- line 492 – please provide the number of sorted microglia cells that were used for the Western blot analysis

For Western blot analysis, we collected 100,000–150,000 Cd11b⁺ cells from each young mouse and 60,000–100,000 Cd11b⁺ cells from each old mouse. We have added the corresponding information in the Methods section (lines 415–416).

- line 608 – it is not clear what was the EdU injection schedule, please rephrase/specify

We apologise for the unclear EdU injection schedule. We have rephrased the schedule in the figure legends and Method section (lines 537–541) and revised Supplementary Figs 4a and 4e.

Supplementary Fig. 4a

Supplementary Fig. 4e

- line 670 – please include these abbreviations in the legend of the relevant figures –figures and their legends should be self-explanatory

Thank you for the suggestion to improve the readability of the manuscript. We have added the abbreviations in the legends of corresponding figures.

Reviewer #3 (Remarks to the Author):

In the manuscript entitled “Comprehensive analysis of the central nervous system revealed microglial senescence and its disease associations” by Matsudaira et al, the authors applied multiple analytical technologies including state-of-the-art single-cell RNA-seq to assess the characteristics of senescent cells in the CNS. They defined microglia (especially DAM) as majority of senescent cells in the CNS white matter. Suppression of microglial senescence by genetic manipulation revealed attenuation of EAE severity. The study is overall very interesting and will provide important new information to the field. However, I have a number of concerns/suggestions that should be addressed/responded to improve the manuscript.

We truly thank you for your appreciation of our work and your valuable suggestions for improving our manuscript.

Concerns/suggested:

1. For scRNA-seq experiment, it is not clear how many mice were used (should be addressed in Figure legend). The results (e.g. Fig. 2D, 3B, 3F, 3G, etc.) should be shown for each mouse in each group. To evaluate and interpret the data it is important to know the biological variation of each feature. In this manuscript, the authors show the results of scRNA-seq only per cell or per group. This makes it impossible for the reader to evaluate the consistency and/or variation of each feature across different animals.

This is indeed a very important and valid point. However, due to the very high cost of the scRNA-seq experiment, we only used one mouse brain per sample to obtain the data, though we were unable to evaluate the biological variation through scRNA-seq analysis. We confirmed the high p16 expression in microglia or disease-associated microglia by independent experiments, including immunofluorescence staining (Figs. 2h-k, 3j-l), Western blot (Fig. 2g and Supplementary Fig. 3c), and RT-qPCR analysis (Figs. 2f, 3i) on the significant number of brains of young and old mice, revealing consistent results across different mice.

2. It's written on Line 410-411: The p16-luc mice and the EAE-induced mice were female, and the other mice were male. What does “the other” refer to? Why were different sexes used in different experiment? Sex of mice used in each experiment should be addressed in the figure legends.

We would like to apologise for the unclear description. We have indicated the sex of the mice

used in each experiment in the figure legends. We previously showed that p16-expressing cells are detected in a similar pattern in both old male and female mice, with the exception of those in reproductive organs (Yamakoshi et al. 2009, PMID: 25923845). In this study, we used male mice to achieve consistent results in most experiments. However, due to the lack of available mice, we used only female mice in experiments where p16-luc mice were used. For EAE experiments, we used only female mice because male mice have been reported to be less susceptible to the EAE phenotype (Voskurl et al. 2011; PMID: 21208397).

3. It's written on Line 415-417: 100 ul of 20 mg/ml Tamoxifen dissolved in corn oil were injected intraperitoneally for five days in a row in Cx3cr1-CreERT2: p16flox/flox mice and then maintained 3-4 weeks before EAE experiments to reduce the effect of p16INK4 depleted lymphocytes such as monocytes and peripheral macrophages⁴⁷ What was the injection schedule of tamoxifen (daily, every two days,...?).

We apologise for the inadequate explanation of the tamoxifen injection schedule. We intraperitoneally injected mice with 2 mg of Tamoxifen daily for 5 days. We have accordingly revised the Methods section (lines 537–541).

4. FACS antibody clones should be addressed.

We have addressed all FACS antibody clones used in this study.

5. It's written on Line 96-98: Further evaluation revealed that p16INK4a mRNA and protein were highly expressed in the brain's white matter regions, such as the corpus callosum and medulla oblongata in old mice (Fig. 1c, d). To increase readability, the authors should decribed in the text which further evaluation they used.

We sincerely thank you for your suggestion to improve readability. We have clarified that these data were obtained by immunoblotting and RT-qPCR. We have accordingly revised the relevant text (lines 107–108).

6. The IHC results shown in Figure 1e should be quantified.

Following the reviewer's comment, we have quantified p16^{INK4a}-positive cells in the corpus callosum of young or old mice. We have included the data in Supplementary Fig. 1f.

Supplementary Fig. 1f

Moreover, we magnified the images and indicated the p16^{INK4a}-positive cells by arrows in Fig. 1e and Supplementary Fig. 1g to highlight the p16^{INK4a}-positive cells.

Fig. 1e

Supplementary Fig. 1g

7. Suppl. Figure 2: cluster-ID should be addressed along with labelling of cell type. The authors may directly label cell type on UMAP. And as mentioned in "1" the results shown in Suppl. Figure 2B should also be visualized per animal, not only per group or per cell.

We appreciate your comments. Per your request, Figs 2b, 2c, 3a, 3b, 4b, and 4c and Supplementary Figs 2a, 7a, 7b, and 7c have been revised. As for the latter part of the comment, the data is derived from only one mouse per sample, as answered above.

Figs. 2b, 2c

Supplementary Figs. 2a

Figs. 3a, 3b

Figs. 4b, 4c

Supplementary Figs. 7a-c

8. In Figure 2, it seems that microglia is the most abundant cell type in the UMAP plot, thus showing only the number of cell expressing Cdkn2a may be misled. The proportion of Cdkn2a-positive cells (e.g. % of total microglia) should also be addressed.

We agree with the reviewer's comment that the proportion of Cdkn2a-positive cells should also be added. As shown in Figs 2b–d and Supplementary Fig 2e, the actual number of Cdkn2a-positive cells was the most prominent in microglia among all cells isolated from the white matter of the brain of mice. In contrast, the high proportions of Cdkn2a-positive cells (expression level > 0) were also observed in neuronal-restricted precursors (NRP, cluster 21) characterised by CDK1 and Sox11 expression (Ximerakis et al. 2019, PMID: 31551601) and T-cells (Supplementary Fig. 2f), although their actual cell numbers were quite limited (23 in NRP and 9 in T cells) compared to microglia (227 cells) (Supplementary Fig. 2e). The results of the p16^{INK4a} expression in the neural progenitor cells in old mice were consistent with those in previous studies (Molofsky et al. 2006, PMID: 16957738; Nicaise et al. 2019, MID: 30910981). We have addressed the result in Supplementary Figs 2e and 2f.

Figs. 2b–d

Supplementary Fig. 2e

Supplementary Fig. 2f

9. Figure 2f: how many mice were used, (semi) quantitative value (per mouse) should be shown.

Per your advice, we have performed semi-quantification for immunoblotting data in Fig 2f. We have added the data in Supplementary Fig 3c.

Supplementary Fig. 3c

10. In some experiments, 33 months old mice were used instead of 22 or 24 months old. This is quite a big difference. Is there any reason of choosing these two different ages for the group of old mice? Would this have any impact on the results?

Since old mice are very difficult to obtain, we have less freedom in the experiments and in choosing the age of old mice. In our results, however, a similar proportion of p16^{INK4a}/Iba1-positive cells was detected in the medulla oblongata of mice at 20 (25.55%, Fig. 4n) and 28 months of age (20.88%, Fig. 2k); no statistically significant difference was observed ($P = 0.1368$). Therefore, in this study, we considered mice older than 18 months as old mice and used 20–33-month-old mice for our experiments. Moreover, we have indicated the age of mice used in each experiment in the figure legends.

11. In Figure 2h-k and other similar figures, it was described “young: $N = 3$, $n > 80$ cells, old: $N = 3$, $n > 200$ cells per mouse”. It would be much better to describe as a range. Also the authors should address how many cells were analyzed (thus e.g. $n = x - y$ out of z cells analysed).

We are grateful for these suggestions. As per your advice, we have addressed the cell number we used for quantification as a range in the figure legends.

12. On Line 547, it was written “6x8 images for medulla oblongata”. What does “6x8 images” mean? How many cells were analyzed in total?

We apologise for the lack of explanation. We used stitched images for quantification (Figs. 1e, 2h, 2j, 3j, 4l, 5e, and 5h, and Supplementary Figs. 4b, 4f, 9b, and 11d); “6x8 images for medulla oblongata” means that we stitched six vertical and eight horizontal images to cover the entire area of the medulla oblongata. After creating stitched images, we enclosed the relevant region and counted all Iba1⁺ cells, p16⁺ cells, EdU⁺ cells, and/or Lgals3⁺ cells in the entire area of each brain region (e.g. Corpus callosum, medulla oblongata, or spinal cord) for quantification. To clarify the method of quantification, we have addressed the cell numbers analysed in the figure legends and

explained the quantification method in detail in the Methods section (lines 472–479).

13. Are there any changes in morphology of aged and/or senescent microglia, compared with young microglia? The authors should also show Iba-1-positive (macrophage/microglia) cells with high magnification to show these possible changes. Also, GF microglia were known to have different morphology compared to SPF microglia. This is impossible to evaluate with low-magnification images (e.g. Figure 4k).

Thank you for this question. Since we are focusing on cellular senescence in microglia, it is worth investigating whether there are morphological differences between p16^{INK4a}-positive senescent microglia/DAM and p16^{INK4a}-negative microglia/DAM under the SPF or GF condition. However, as mentioned in the response to reviewer #2, high-sensitivity p16^{INK4a} staining is not compatible with the preservation of microglial morphology. Our main focus in this study was on function rather than morphology in senescent microglia, but the morphological difference between p16^{INK4a}-positive senescent microglia/DAM and non-senescent microglia/DAM should be investigated in the future.

14. Suppl. Figure 10: another method should be used to validate the model (i.e. expression of p16INK4a), for qPCR or IHC.

Thank you for your comment on the Cx3cr1-CreER^{T2}:p16^{INK4a} flox/flox mouse model. We have further investigated the deletion efficiency of p16^{INK4a} in Iba1-positive cells using a cuprizone diet, which promotes the acute microglial proliferation in the corpus callosum due to demyelination (Praet et al. 2014; PMID: 25445182). Tamoxifen injection suppressed the increase in the number of p16^{INK4a}-positive cells that accompanied the increase in the number of Iba1-positive cells, although some p16^{INK4a}-positive cells were detected after tamoxifen injection in agreement with the immunoblotting data in Supplementary Fig. 11b. We have added the data in Supplementary Figs. 11c-e.

Supplementary Figs. 11c-e

15. The differences between results shown in Figure 5K and 5P are quite interesting, the authors should discuss more about this, e.g. involvement of other cell types, insufficiency of know out, etc.

We are very grateful for this comment. We have already mentioned that the knockout efficiency is insufficient in this mouse line (Supplementary Figs. 11b-e). However, it is tempting to speculate other possibilities, including the involvement of other cell types. We have more clearly discussed this matter in lines 295–304.

16. Suppl. Figure 9B-G: proportion (e.g. % of CD45+) should be shown along with absolute cell number.

Thank you for this suggestion. We have added the data showing the proportion of Cd45⁺ cells (Supplementary Figs. 10 h–k). We found that, in addition to the decrease in absolute cell number (Supplementary Fig. 10c), the proportion of Cd4⁺ T-cells in Cd45⁺ cells in the p16 KO mice in which EAE was induced has also decreased (Supplementary Fig. 10i), suggesting that the amelioration of the EAE phenotype may be partially due to the suppression of recruitment of T-cells in the spinal cord.

Supplementary Figs. 10 h–k

17. At the end, the link between microglial senescence in aged mice and those in pathology that occur in young mice is unclear. Also, although in young mice the microglial senescence is hardly detected, suppression of p16INK4a in young mice could provide significant effects in EAE. This needs some more discussion. Would this possibly mean that the disease causes microglial senescence and not another way around, but how suppression of this cascade can give positive effects on disease severity?

We showed that the number of senescent microglia decreases in old GF mice (Figs. 4h, 4l, and 4n). In terms of the impact of resident bacteria on the pathogenesis of CNS, a mouse model of AD (5x familial AD) has also been reported to exhibit reduced A β depositions and attenuated neuronal loss under germ-free (GF) conditions (Mezö et al. 2020, PMID: 32727612; Erny, et al. 2021, PMID: 34731656). Given that AD model mice reportedly show senescent-like signatures in microglia (Hu et al. 2021, PMID: 34107254; Brelstaff et al. 2021, PMID: 34669475), it is tempting to speculate that bacteria induce cellular senescence in microglia during ageing, as well as under pathological conditions.

As indicated, microglial senescence is hardly detected in young normal mice. However, we detected microglial senescence in the spinal cord of young mice in which EAE was induced (Figs. 5h-j). We also showed that the suppression of p16^{INK4a} in young mice attenuated demyelination and limb paralysis (Figs. 5k-r). Importantly, the impact of p16^{INK4a} deletion in Cx3cr1-CreER^{T2} p16^{flox/flox} mice lasts a lifetime after tamoxifen administration. Therefore, we suppose that microglial senescence promoted the disease phenotype in the EAE model, although the mechanisms need to be elucidated in future studies.

18. In the Methods section, an irradiation experiment was described. This experiment seems to be missing or unclearly described.

We are sorry for unclear description. We addressed the irradiation experiment in Supplementary Figs. 1e and 11b. We have added relevant information to clarify all corresponding aspects in the Methods section (lines 350–357).

REVIEWERS' COMMENTS:

Reviewer #1 (Remarks to the Author):

The authors have addressed my comments satisfactorily, and I have no further issues to raise.

Reviewer #2 (Remarks to the Author):

The authors have adequately addressed the comments and recommendations of the reviewers.

Reviewer #3 (Remarks to the Author):

The authors have revised the manuscript and updated the figures and figure legends to include additional information.

However, as I have commented before, I would strongly recommend to address in the figure legend the number of animal the authors used for single-cell sequencing. It is critical to build up the hypothesis on the basis of "n=1" experiment. However, the authors may discuss about this point and about the consistency of their findings using other methods, in order to support their hypothesis. Alternatively, to date, there are numerous available open source data that the authors may use to support their findings. It is very important to be transparent. This will help readers to interpret and estimate the work.

Point-by-point responses to the reviewers' comments

We would like to reiterate our sincere thanks to the reviewers and the editor for their careful consideration. We are very pleased that we have been able to address most of the reviewers' concerns. Our point-by-point responses to the reviewers' comments are given below, with the original comments in bold.

Reviewer #1 (Remarks to the Author):

The authors have addressed my comments satisfactorily, and I have no further issues to raise.

We are very grateful for your comments on the improvement of our manuscript.

Reviewer #2 (Remarks to the Author):

The authors have adequately addressed the comments and recommendations of the reviewers.

Your comments are quite helpful in improving our manuscript.

Reviewer #3 (Remarks to the Author):

The authors have revised the manuscript and updated the figures and figure legends to include additional information.

However, as I have commented before, I would strongly recommend to address in the figure legend the number of animal the authors used for single-cell sequencing. It is critical to build up the hypothesis on the basis of "n=1" experiment. However, the authors may discuss about this point and about the consistency of their findings using other methods, in order to support their hypothesis. Alternatively, to date, there are numerous available open source data that the authors may use to support their findings. It is very important to be transparent. This will help readers to interpret and estimate the work.

We are very grateful to you for this comment, which is an emphasis on the sample size of our scRNA-seq data. To clarify this point for readers, we have added the information that our scRNA-

seq is based on an N = 1 experiment in the figure legends of the main and supplementary figures. In addition, we have added the following cyan highlighted sentences in the Discussion section (lines 267–270).

Although we only analysed one mouse brain per group in our scRNA-seq study, the induction of senescence in microglia was consistently confirmed in various experiments using a significant number of brains from young and old mice (Fig. 2f-k, 3i-l, 4l, 4n-p, Supplementary Fig. 3c).